# Spatial analysis with SPIAT and spaSim to characterize and simulate tissue microenvironments

Yuzhou Feng[1], Tianpei Yang[1], John Zhu [1], Mabel Li[1], Maria Doyle[1], Volkan Ozcoban [1], Greg T. Bass[2], Angela Pizzolla [1,3], Lachlan Cain[1], Sirui Weng[1,3], Anupama Pasam[1], Nikolce Kocovski[1], Yu-Kuan Huang[1,3], Simon P. Keam [1,3], Terence P. Speed [4], Paul J. Neeson [1,3], Richard B. Pearson [1,3], Shahneen Sandhu[1,3], David L. Goode[1,3] & Anna S. Trigos [1,3] ✉

Spatial proteomics technologies have revealed an underappreciated link between the location of cells in tissue microenvironments and the underlying biology and clinical features, but there is significant lag in the development of downstream analysis methods and benchmarking tools. Here we present SPIAT (spatial image analysis of tissues), a spatial-platform agnostic toolkit with a suite of spatial analysis algorithms, and spaSim (spatial simulator), a simulator of tissue spatial data. SPIAT includes multiple colocalization, neighborhood and spatial heterogeneity metrics to characterize the spatial patterns of cells. Ten spatial metrics of SPIAT are benchmarked using simulated data generated with spaSim. We show how SPIAT can uncover cancer immune subtypes correlated with prognosis in cancer and characterize cell dysfunction in diabetes. Our results suggest SPIAT and spaSim as useful tools for quantifying spatial patterns, identifying and validating correlates of clinical outcomes and supporting method development.

Recent advances in spatial multiplex proteomics technologies, such as OPAL multiplex immunohistochemistry (mIHC), multiplexed ion beam imaging (MIBI)[1], co-detection by indexing (CODEX)[2], and image mass cytometry (IMC)[3] allow detailed cell phenotype profiling by simultaneously detecting up to 80 proteins in tissue sections. These technologies can generate multidimensional datasets consisting of location coordinates, protein expression intensities, and morphological features of hundreds of thousands of cells.

Spatial technologies have revolutionized how we study the immune microenvironment, especially in cancer. Spatial proteomics technologies have revealed an immense complexity of spatial patterns of immune cells linked with prognosis and response to immune checkpoint inhibitors in several solid tumors[4–12]. These platforms have

now become an essential tool to help interrogate the complexity of tumor immunobiology and are positioned to contribute to a new generation of biomarkers for tumor classification and treatment selection. Beyond cancer, spatial proteomics technologies have been applied across fields, such as helping to characterize the evolution of islets and their immune neighborhood during type I diabetes progression[13] and understand the structure and complex immunoregulation in tuberculosis granulomas[14]. Applications have not been limited to human samples, as these platforms have also been used to characterize the immune responses to Ebola infection in Rhesus Macaque[15] and the composition of the mouse spleen and atherosclerotic plaque[16,17]. The range of applications is set to further increase as these platforms become more accessible.

[1]Peter MacCallum Cancer Centre, Melbourne, VIC, Australia. [2]Research & Development, CSL Innovation, Parkville, VIC, Australia. [3]The Sir Peter MacCallum Department of Oncology, The University of Melbourne, Melbourne, VIC, Australia. [4]Bioinformatics Division, The Walter and Eliza Hall Institute of Medical Research, Parkville, VIC, Australia. ✉e-mail: anna.trigos@petermac.org

Unfortunately, spatial analysis methods have lagged behind technological advancements. To date, most method development efforts have focused on extracting information from raw microscopy images through cell segmentation and cell phenotyping. However, once these have been captured, there are few downstream methods to quantify spatial data, and these mainly focus on simplistic basic analysis, such as distances between pairs of cells. Furthermore, these methods often rely on arbitrary thresholds, significant user input, and focus on simple dominant patterns. As a result, they fail to capture the diversity of spatial patterns present and address the complex biological questions that often underpin spatial analysis. Adding to this, computational tools for tissue spatial data analysis are generally tailored to specific platforms[18], focus on the engineering of creating an infrastructure for spatial data, which often translates as being most suited for users with significant computational ability[19–21], or were not developed for data generated from tissues[22], limiting their usability.

The development of spatial analysis methods has been further hampered by a lack of simulators of spatial data to enable robust, reproducible, and quantitative assessment of new approaches. Current point pattern simulators, such as those available in spatstat[22] do not reflect the inherent complexity or common patterns observed in biological tissues. As a result, we are currently unable to benchmark the ability of individual methods to capture and characterize distinct spatial patterns.

To aid in the advancement of spatial analysis methods commensurate with the development of new spatial platforms and the diversity of tissues that are beginning to be profiled, here we present SPIAT (spatial image analysis of tissues), a platform-agnostic, user-friendly analysis toolkit in R for the characterization of spatial patterns of multiplex tissue images, as well as spaSim (spatial simulator), a novel simulator of tissue cell spatial patterns for benchmarking of spatial metrics. With case studies in prostate, breast, colon cancer, melanoma, and diabetes, we identify immune subtypes associated with patient survival and improve our understanding of cell dysfunction in type I diabetes. We envision SPIAT and spaSim will aid the research community's efforts in developing, standardizing, and testing spatial analysis methods while empowering novice users in their spatial analysis journey, helping push forward the use of spatial metrics in basic, translational, and clinical research.

## Results

### Overview of SPIAT and spaSim

SPIAT is a spatial analysis toolkit implemented in R compatible with data from any spatial technology that generates a table of cell coordinates and cell phenotypes, and/or marker intensities, which are the required input for SPIAT. Methods in SPIAT are based on deterministic algorithms, spatial statistics, and mathematical equations that derive metrics of particular spatial features from individual images. Across six analysis modules and over 40 functions, SPIAT includes visualization, distance-based analyses, colocalization metrics, automated algorithms for the detection of structures and structure margins without user input, classification of cell-cell relationships without the use of thresholds and multiple algorithms for the identification of cellular neighborhoods (Fig. 1). SPIAT can also quantify the heterogeneity of tissue microenvironments within a single tissue section, which to date has been a largely underappreciated aspect of spatial analysis. SPIAT has a series of quality control steps that allows the detection of staining artifacts, cell phenotyping, and the exclusion of incorrectly phenotyped cells (Note N1). SPIAT significantly enhances our ability to perform comprehensive spatial analysis of tissue microenvironments (Supplementary Table I). Implementations of the algorithms were made to optimize speed, making it capable of analyzing at least 1 million cells on a local computer in under 30 min.

spaSim is a first-of-its-kind simulator of tissue spatial data implemented in R (Fig. 2). The purpose of spaSim is the testing of spatial metrics in a clean and controlled environment to understand their behavior across different ranges of spatial patterns generated with different parameter settings. In spaSim, simulated images are constructed in a stepwise fashion, starting from a background of cells simulated with either a Hardcore process for tumor tissues or an evenly spaced model for normal tissues, followed by assigning cell types using random number sampling and the use of geometric shapes (Fig. 2 and Supplementary Fig. 1). The simulated images recapitulate the main spatial properties of tissue sections, including the overall distribution of background cells, tissue regions, clusters, infiltration and exclusion of cell types from an area, and blood/lymphatic vessels, allowing more representative simulations than with point pattern-based packages (Notes N2, N3, and Supplementary Table II). spaSim can also simulate images that reproduce the spatial properties of real tissue images after extraction of basic features, such as the total number of cells, cell type proportions, and the number, size, and location of clusters (Note N3). To simplify benchmarking, spaSim allows the simulation of a series of images varying by any parameter specified by the user.

### SPIAT and spaSim to measure and benchmark cell colocalization metrics

Cell colocalization methods have been popular in providing an overall score representing the closeness between pairs of cell types, and are often intuitively interpreted as a measure of cell interaction. However, this interpretation is often an oversimplification as cells can adopt a diversity of spatial configurations with often distinct biological and clinical interpretation. For example, in the case of tumor microenvironments, lymphocytes can infiltrate tumor areas, which has been associated with better control of tumor growth and improved prognosis[7]. Immune cells remaining distal to the tumor area (stromal immune cells), such as those forming tertiary lymphoid structures (TLS), are linked to response to immune checkpoint inhibitors[9]. In contrast, immune-excluded tumours, including tumors with immune cells sequestered to the tumor margin, forming an 'immune ring', are less likely to show an association with response to immune checkpoint inhibitors[23,24]. Unfortunately, we have no clear understanding of the ability of colocalization metrics to capture each of these patterns.

SPIAT includes several metrics to measure colocalization, including the average pairwise distance (APD), the average minimum distance (AMD) between cells, the cells in the neighborhood (CIN)[25], the cell mixing score (MS)[5] with a novel normalization method to account for cell number (normalized mixing score−NMS) (Methods, Supplementary Fig. 2), the cross K function and the calculation of the AUC (Fig. 3a). To measure their ability to capture clinically relevant patterns, we simulated images using spaSim across a range of different spatial patterns commonly encountered in tumor microenvironments. These ranged from a cluster of reference tumor cells with a stromal cluster of target immune cells at different distances from the reference tumor cluster, the formation of immune-cell rings surrounding the tumor cluster, to various levels of infiltration (Fig. 3b). We determined the ability of each metric to capture these patterns (i.e. its performance) as the change in the score when a pattern was present. A summary of the results is presented in Supplementary Table III.

The APD and AMD performed similarly across simulations, where scores decreased with higher levels of infiltration, while the NMS and the Cross K AUC followed the opposite trend. It was possible to distinguish immune cells in the stroma from those infiltrated in the tumor cluster based on the scores generated by these four methods, but not the degree of infiltration. APD and AMD were able to capture the tumor-stromal immune distance, whereas neither the NMS nor the Cross K AUC was sensitive to the distance between tumor clusters and immune clusters. CIN and MS could also distinguish infiltrated immune cells from stromal immune cells. These metrics could detect the levels of immune infiltration, but not the tumor-stromal immune distance.

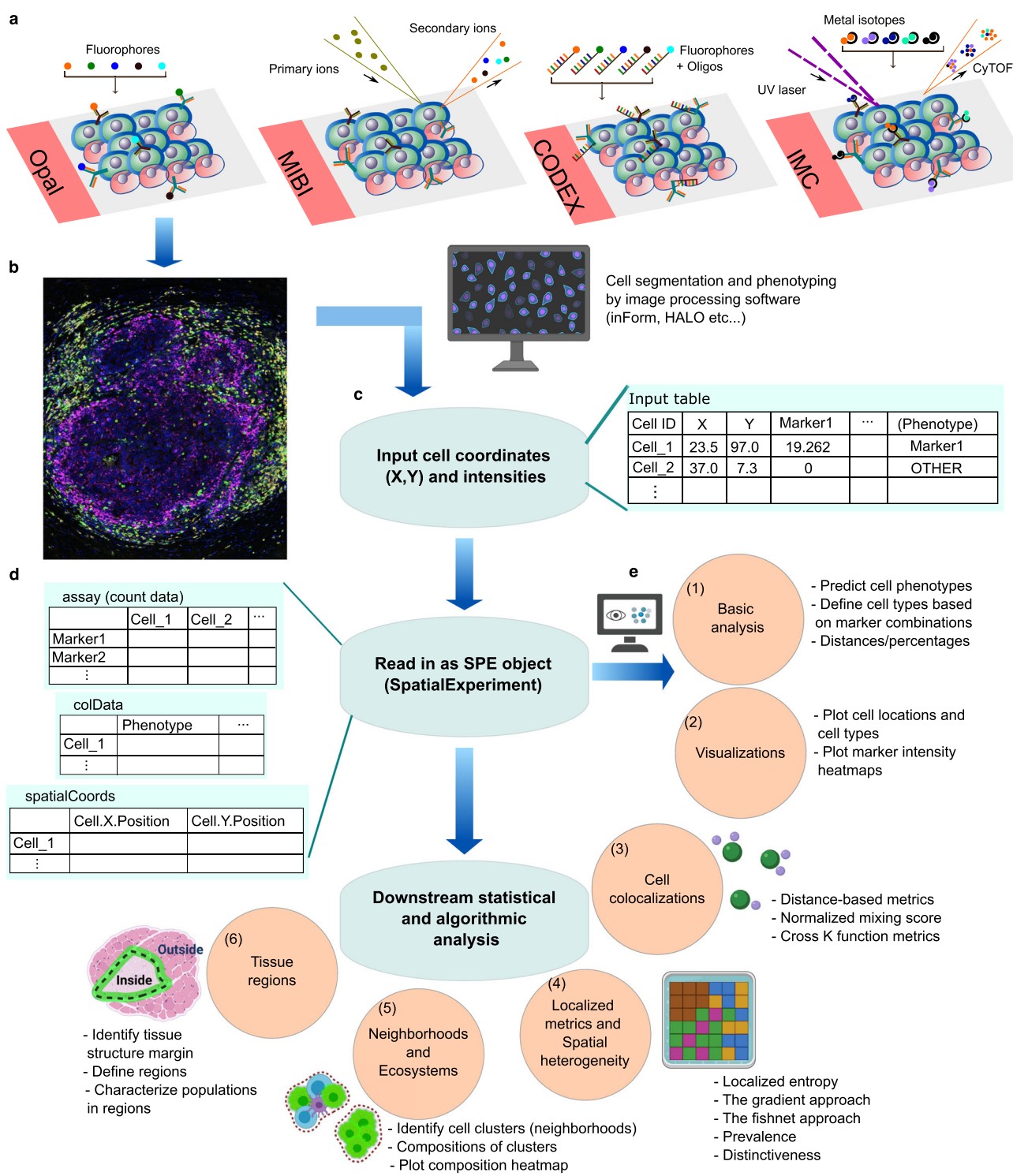

**Fig. 1 | Overview of SPIAT analysis modules. a** Examples of technologies used to generate spatial proteomics data. **b** These platforms generate multiplex images, which are then processed by image analysis software to perform cell segmentation and marker deconvolution. **c** This results in data in the form of a table with cell coordinates and cell phenotypes and/or marker intensities that can be exported from the image analysis software. **d** SPIAT begins by reading in this table of cell IDs, X, Y coordinates, cell phenotypes (if available), and marker intensities (if available). Any other additional columns of data are optional and not required. Any data format that can be read into R, with cell coordinates as vectors and matched cell phenotypes and/or marker intensities, can be used as input to SPIAT. This table is then converted to a SpatialExperiment object by SPIAT, a widely used format for spatial data in R. **e** This is followed by analysis by one of six analysis modules. Users can predict cell phenotypes or define cell types directly as well as perform basic quality control and calculate basic metrics such as cell percentages and cell distances (1), followed by visualization of the spatial distribution of cells (2), quantification of cell colocalization through seven distinct methods (3), measure spatial heterogeneity (4), detect neighborhoods or ecosystems of cells based on clustering and community detection (5), and characterize cell types relative to the margin of tissue structures, such as the tumor margin (6). Created with BioRender.com. SPE spatial experiment object.

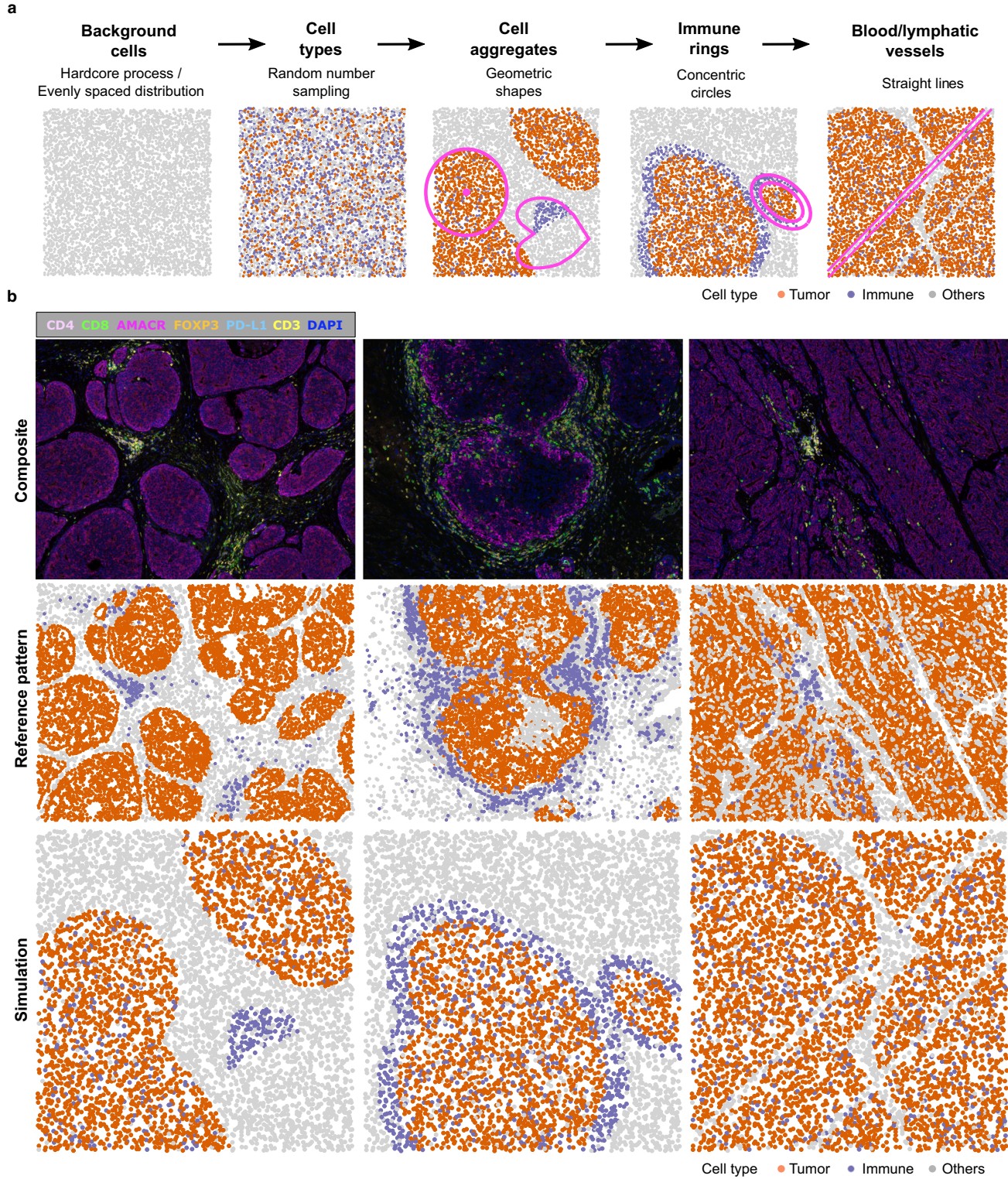

**Fig. 2 | spaSim simulator of spatial data from tissues. a** Steps in simulation. After simulating the background cells, the rest of the steps are optional and can be mixed and matched as desired. **b** Examples of patterns in prostate cancer tissue images, shown as composite microscopy images and after computer-rendering by SPIAT, followed by the simulation of the observed spatial patterns using spaSim. spaSim allows the simulation of the main tumor and immune patterns found in tumor tissues. Source data are provided as a Source Data file.

Furthermore, CIN and MS gave similar scores to cases of immune ring formation and immune infiltration, suggesting they could not distinguish these.

Since no method could differentiate the formation of immune rings from infiltration or stromal immune cells, we developed a new metric based on the crossing of the observed and expected curves in the cross K function (Cross K intersection− CKI) (Fig. 3a, bottom right). The CKI was higher when immune cells were close to or bordering the tumor area (Fig. 3b, bottom row, blue). Further analysis revealed that the CKI performs best with clearer tissue structures of the reference

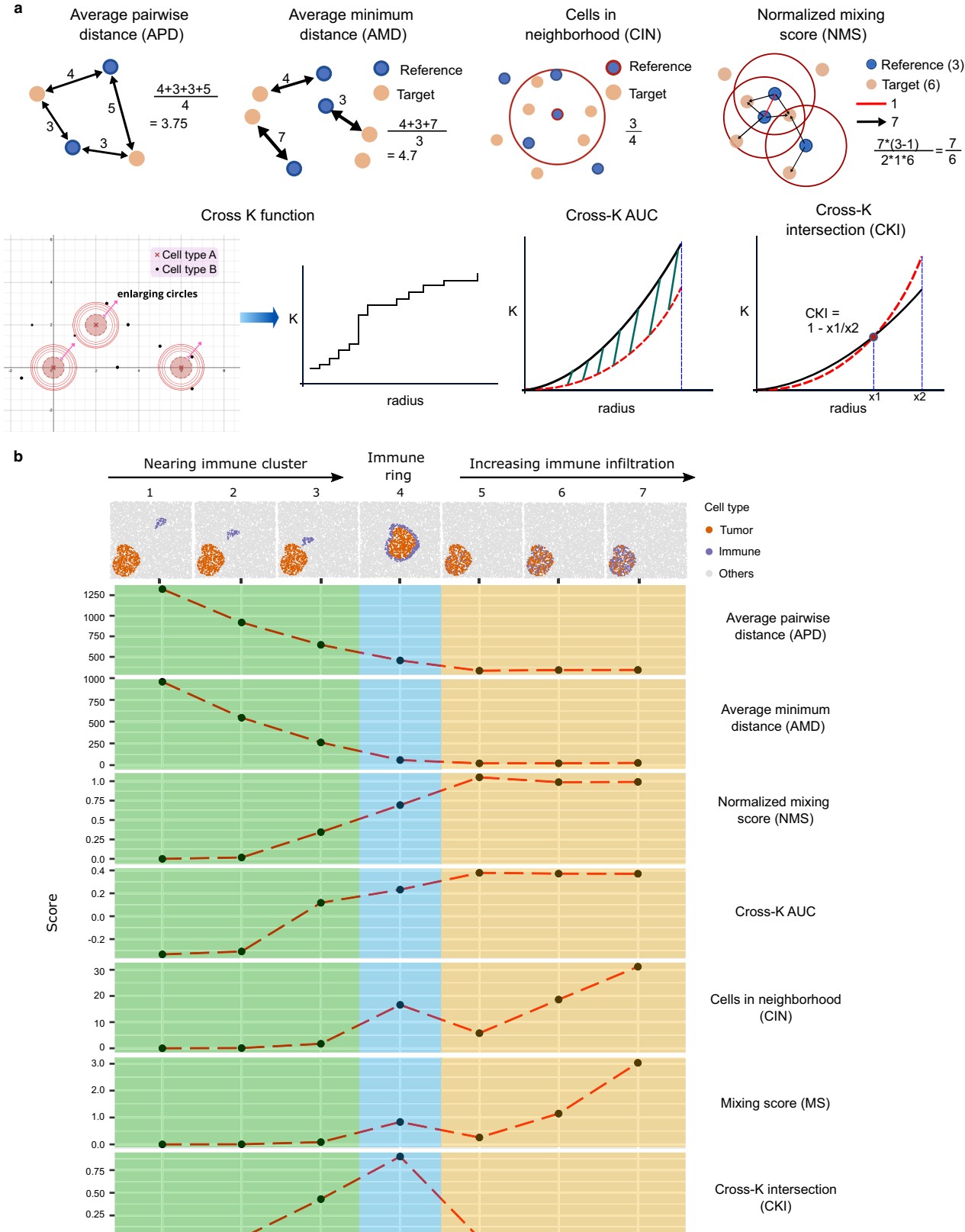

**Fig. 3 | Cell colocalization metrics. a** Diagrams of the cell colocalization metrics available in SPIAT. **b** Simulations of common tumor-immune spatial patterns and the scores generated by each metric. For the MS, NMS, and CIN, we chose 500 as the distance parameter. For the cross K function-related metrics, we chose 0–500 as the range of the radii. Simulations were generated with spaSim. APD average pairwise distance, AMD average minimum distance, CIN cells in the neighborhood, MS mixing score, NMS normalized mixing score, AUC area under the curve, CKI cross K intersection. Source data are provided as a Source Data file.

cell types (e.g. tumor areas) and with a meaningful number of target cells (e.g. immune cells) present (Supplementary Fig. 3).

Our results suggest that metric selection should be guided by the underlying spatial pattern one aims to capture. A combination of complementary metrics is likely needed to capture and quantify distinct patterns.

## Entropy gradients for tissue classification

Key limitations in the application of scores to characterize cell colocalization is the need to set arbitrary thresholds (e.g. for the minimum distance for interaction) and the reliance on relative metrics that generally can only be interpreted within a cohort. This is particularly problematic in the translation of spatial patterns and cell colocalization metrics as biomarkers of prognosis and treatment response, which often require the classification of samples into groups.

We have developed the novel concept of entropy gradients as a self-contained metric to define the attraction and repulsion of cell types. The calculation of entropy depends on the relative proportion of one cell type to other cell types of interest in an area. If the proportions of all the cell types in an area are the same, then the entropy is at its maximum. If one of the populations decreases its proportion, becoming rarer than the others, the entropy decreases. Similarly, if one of the populations increases its density, becoming more common than the others, the entropy also decreases. Hence, entropy measures how distinct the proportions of different cell types are in an area.

In entropy gradients, SPIAT identifies concentric circles with a range of radii centered around each cell of the reference population (Fig. 4a). It next identifies the reference and target cells within the circles and calculates the aggregated entropy (Methods), resulting in a single score per image for each concentric circle. Finally, resulting entropies are plotted against the corresponding radii. In cases of attraction between cell types, scores are higher close to the cells of the reference population and decrease as the distance increases, resulting in a negative slope (or remain unchanged if the attraction occurs throughout the image) (Fig. 4b). In contrast, a positive slope would be observed for repulsion (Fig. 4b). Since the definition of attraction and repulsion is based on the slope of the curve, the application of Entropy Gradients enables sample classification independent of other samples and without thresholding.

We generated a set of simulated images with spaSim with various levels of colocalization between two cell types (Fig. 4c, top row). Here, immune cells were used as the reference population (see Note N4 about the choice of reference population). Cases of infiltration of immune cells to areas with a high density of tumor cells led to strong negative slopes, indicating attraction (Fig. 4c, Images 1, 2). In contrast, the presence of stromal immune cells was associated with a strong positive slope, indicating repulsion (Image 3). In cases of immune cells forming a ring around an aggregate of tumor cells (Image 4), the gradient pattern indicates first repulsion, followed by attraction, consistent with attraction only in the outer areas of the region of tumor cells.

To demonstrate the power of entropy gradients, we investigated a cohort of 40 triple-negative breast cancer samples profiled with MIBI[5] (Fig. 4d and Supplementary Fig. 4). We applied our Entropy Gradients method with each of the immune populations as the reference, generating one curve per sample. We found a major distinction between samples with attraction and repulsion of immune cell types (Fig. 4e). Visual inspection of the images corroborated the classification (Fig. 4d). Using entropy gradients, we discovered that attraction of macrophages, cytotoxic and helper T cells with tumor cells was associated with a longer time to death (i.e., prolonged survival) (Fig. 4f), which was not reported in the original study[4].

This case study demonstrates the application of entropy gradients as a self-contained approach to classify samples based on spatial patterns, elucidating novel associations with patient outcomes in breast cancer.

## Spatial heterogeneity to understand the distribution of patterns

Cell colocalization metrics are designed to capture dominant spatial patterns, generating a score per image representing how strongly 'on average' a pattern is represented. However, tissues are highly heterogeneous, often with multiple tissue structures present, with further heterogeneity in the stromal populations. As a result, patterns are often unevenly distributed through a tissue, and multiple distinct patterns can co-exist. To measure this, SPIAT splits images into grids using a fishnet approach (Fig. 5a). This is followed by calculation of the localized entropy, which captures the diversity of cell types present in each grid square (Supplementary Fig. 5). Finally, we assess the Prevalence of a pattern by quantifying the percentage of grid squares positive for the pattern, and the Distinctiveness, which captures whether a pattern is spread across an image or confined to particular areas using spatial autocorrelation (Fig. 5a).

We simulated images with a mixture of spatial patterns of two cell types, Cell A and Cell B (Fig. 5b, middle row) and aimed to investigate the spatial heterogeneity of the co-occurrence of these two cell types. As expected, images with vast areas of co-occurrence showed higher Prevalence scores, but cases where this pattern was confined to a particular area of the image had higher Distinctiveness scores (Fig. 5b, top and bottom row, respectively). We also performed this analysis in the cohort of TNBC[5] (Supplementary Fig. 4), with the pattern of interest being the colocalization of tumor cells and macrophages. As in Fig. 5b, these scores were complementary (Fig. 5c). Patients 16 and 4 had similar Prevalence scores with the colocalization of tumor cells and macrophages covering a similar proportion in both samples. However, Patient 16 had a lower Distinctiveness score as the colocalization squares are spread out throughout the sample, whereas in Patient 4, they were aggregated in a specific area. In contrast, Patients 13 and 24 have similar Distinctiveness scores as the colocalization squares are both widespread in the images. However, Patient 13 has a higher Prevalence score showing more areas with high colocalization of tumor and macrophage cells. This pattern was randomly distributed in the tissue sections of Patients 16, 13, and 24, as their Distinctiveness was close to zero.

Spatial heterogeneity allows us to go beyond identifying dominant spatial patterns based on the interactions of pairs of cells, to now capturing the spatial distribution of specific patterns of interest in a tissue section.

## Classifying regions relative to a tissue structure

An alternative approach to capture the spatial heterogeneity of patterns is the classification of cells into regions relative to the tissue structures present. A common example is describing the location of immune cells relative to the tumor margin. To date, these approaches often rely on qualitative judgements for classification, where users manually draw the outlines of structures (margins) or annotate training sets for machine learning, which are both time-consuming and subject to bias.

SPIAT allows the automated detection of tissue structures and margins without the need to manually annotate regions or train a classifier, improving reproducibility and reducing analysis time. This is followed by classification of cell populations as stromal, infiltrated, or forming a ring in the inner or outer margin, bordering the structure (Fig. 6a). SPIAT returns the proportions of cell types at each of these regions and their distance to the margin (Supplementary Fig. 6 and Note N5).

Since some tissues do not have clear margins, we developed the Ratio of Bordering cell count to Cluster cell count (R-BC) metric, where lower values correspond to samples where a true margin is more likely to exist (Supplementary Fig. 6). This approach was tested

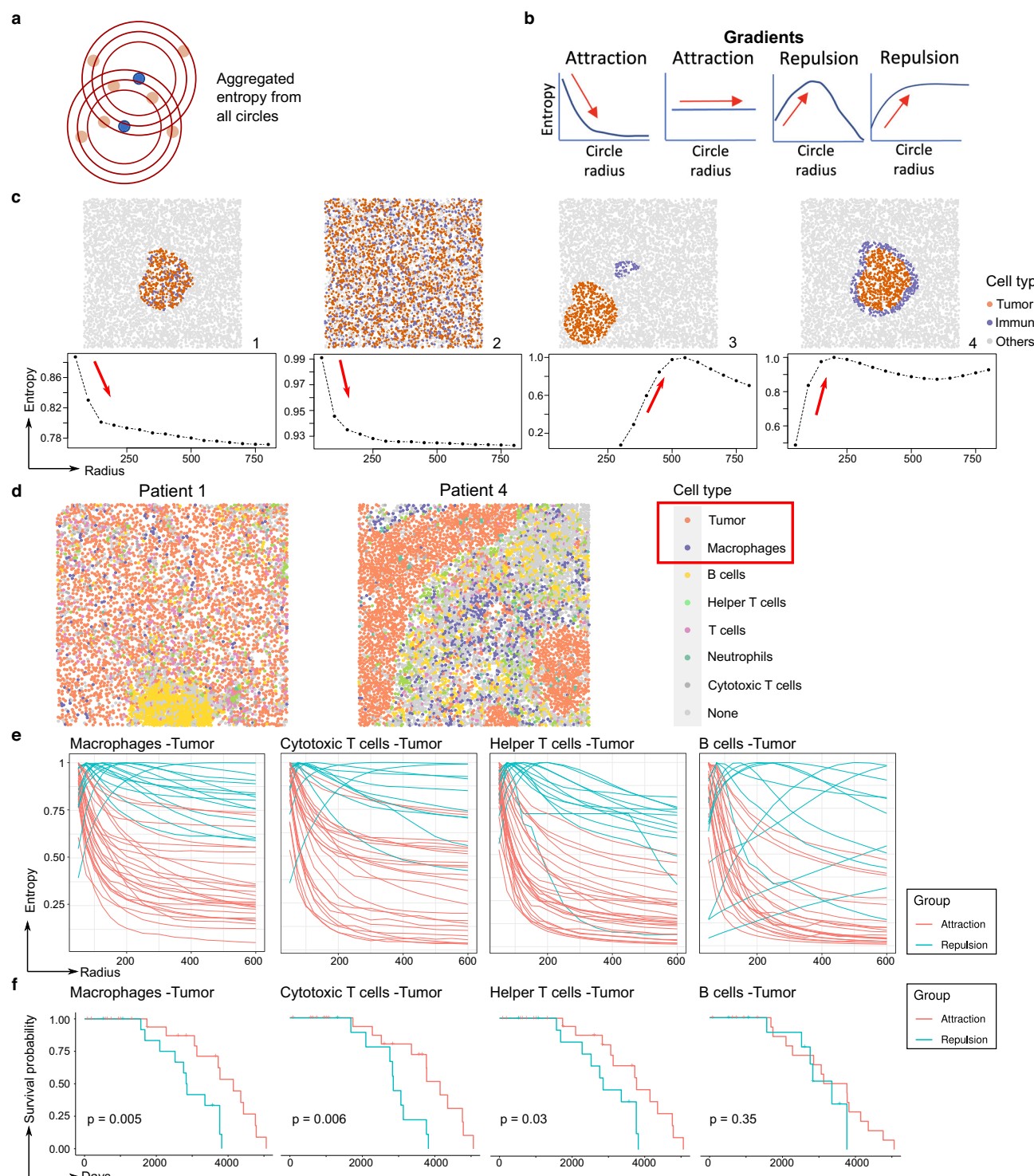

**Fig. 4 | Entropy gradients for self-contained sample classification. a** Entropy is calculated from concentric circles around cells from a reference cell population. **b** Entropy scores are then displayed as a gradient along a radius. The shape of the entropy gradient curves is used to classify patterns as attraction or repulsion. **c** Simulations of different levels of colocalization between immune and tumor populations and the resulting entropy gradient curves. The slope of the curve allows a self-contained classification of samples. **d** Examples of the spatial distribution of immune cells in TNBC samples, showing either attraction of tumor cells and macrophages (Patient 1) or repulsion between these cell types (Patient 4). We used SPIAT to predict the phenotypes de novo based on the marker intensities of CD3, CD4, CD8, CD20, MPO, and CD68 to identify helper and cytotoxic T cells, B cells, neutrophils, and macrophages. **e** Entropy gradients of macrophages,

cytotoxic T cells, helper T cells, and B cells with tumor cells in the TNBC MIBI dataset. Each curve corresponds to a sample. Samples were classified based on their initial slope (near zero radius) with negative (attraction) and positive (repulsion) slopes. *n* = 40 samples from 39 patients. Samples without the relevant cell types were excluded from the particular plot. **f** Kaplan–Meier survival analysis of the classified samples. The time shown corresponds to days to death. A significant association was found for the interactions involving macrophages and cytotoxic and helper T cells, but not B cells. *p* values calculated using a two-sided log-rank test. No adjustment for multiple comparisons was performed. *n* = 36, 31, 35, and 30 patients, respectively. Patients with samples without the relevant cell types were excluded from the survival analysis. Source data are provided as a Source Data file.

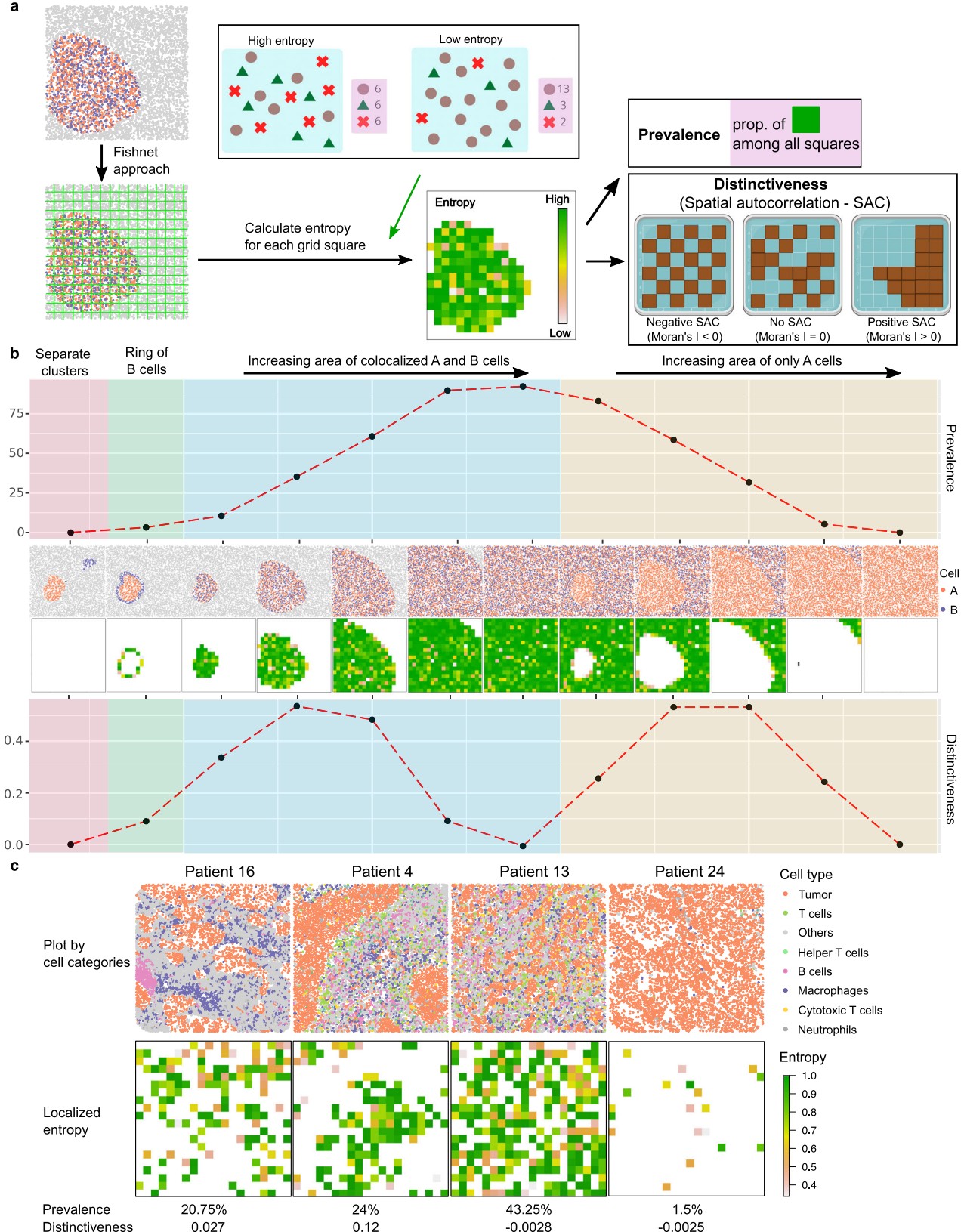

**Fig. 5 | Spatial heterogeneity metrics in SPIAT. a** Calculation of spatial heterogeneity of a pattern. A fishnet is used to split an image into grid squares. Next, the pattern of interest is quantified in each grid square. The Prevalence of a pattern is the percentage of grid squares positive for the pattern, and the Distinctiveness refers to how common the pattern is in the image, calculated by global spatial autocorrelation. Triangles, circles and crosses represent different cell types. **b** Calculation of the Prevalence and Distinctiveness of a pattern of co-occurring

A and B cell populations measured using entropy in a set of simulated images. Note that the two metrics do not mirror each other. Images with an even spread of A and B cells have a high Prevalence but low Distinctiveness. Images where a pattern is found in confined areas tended to have higher Distinctiveness scores and lower Prevalence scores. **c** Prevalence and Distinctiveness scores in the TNBC cohort of a pattern of colocalization of tumor cells and macrophages. Source data are provided as a Source Data file. Created with BioRender.com.

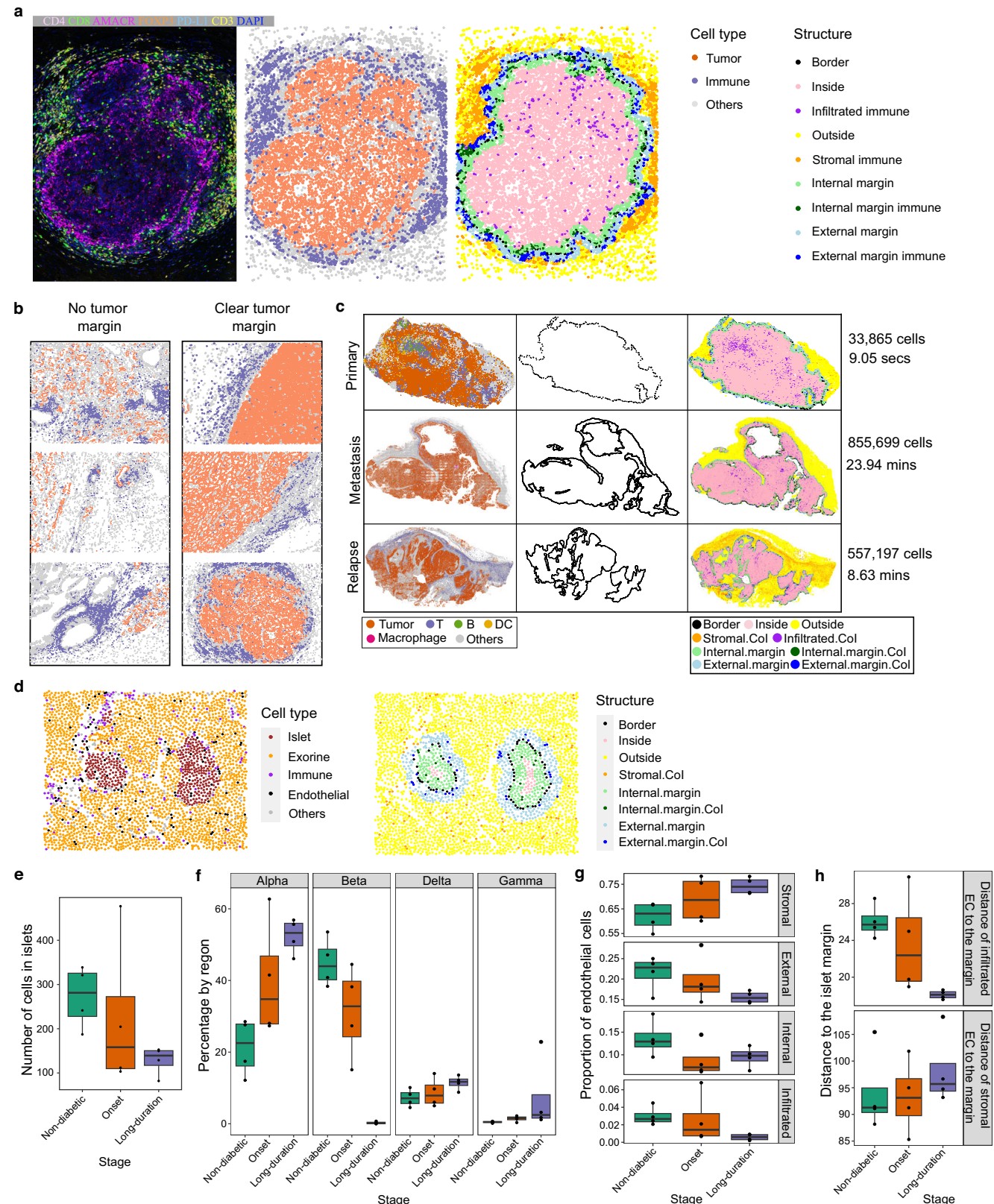

in a set of prostate cancer tissue sections, allowing us to distinguish samples with and without clear tumor margins (Fig. 6b and Supplementary Figs. 7, 8). Furthermore, we tested our margin detection and tissue structure classification in three large whole-tissue section images from a melanoma patient with 33,865, 855,699, and 557,197 cells each (Fig. 6c, left column and Supplementary Fig. 9). Our margin

detection algorithm accurately identified the tumor margins despite the myriad of tumor regions and complex shapes (Fig. 6c, middle column). The subsequent classification of immune cells was also intuitive (Fig. 6c, right column), which was then used to derive the proportion of cell types in each region and calculate their distance to the tumor margin (Note N5).

**Fig. 6 | Automated classification of cells relative to a tissue structure margin performed by SPIAT. a** Tumor margin detection and immune cell classification relative to tumor margin in a prostate cancer tissue. **b** Prostate cancer tissues with and without clear tumor margins based on R-BC scores. Shown are the top three and bottom three images with the highest and lowest scores, with a minimum of 300 tumor cells and 300 immune cells. Color codes of panel **a** were used. **c** Tumor margin detection and immune cell classification in melanoma. The time displayed corresponds to runtime on a local computer (16 GB RAM, 8-core CPU, Apple M1 Pro Chip). Highlighted cells of interest (CoI) are T cells, B cells and DCs for primary and relapse samples, and macrophages, T cells, B cells and DCs for the metastasis sample. **d** Pancreatic tissue islet of a patient at the onset of diabetes, showing individual cell types and the detection of islet margin and areas by SPIAT. **e** Islet size reduced during disease progression. Decreasing one-sided JT $p$ value = 0.033. **f** Percentage of islet cell types in islets. The percentage of beta cells decreases as the disease progresses, whereas other cell types increase. Decreasing one-sided JT $p$ value = $6.40 \times 10^{-4}$ for beta cells. Increasing one-sided JT $p$ values = $6.15 \times 10^{-3}$, 0.063, $9.93 \times 10^{-3}$ for alpha, delta, and gamma islet cell types, respectively. **g** Distribution of EC relative to islet structures. The proportion of EC in the stroma increases during progression (increasing one-sided JT $p$ value = 0.023), but becomes depleted within islets and in the external margins of islets (decreasing one-sided JT $p$ value = 0.015 and 0.03, respectively). The internal margin showed no trend (one-sided JT $p$ value = 0.11). **h** Distance of infiltrated and stromal EC to islet margins. Infiltrated EC localize closer to the islet margin during progression (decreasing one-sided JT $p$ value = 0.0020), but stromal EC become further away from islet margins (increasing one-sided JT $p$ value = 0.14). **e–h** $n$ = 12 biologically independent samples. In boxplots, the center line corresponds to the median and the box limits correspond to the first and third quartiles (the 25th and 75th percentiles). The upper and lower whiskers extend to the maximum or minimum value within 1.5 times the interquartile range, respectively. CoI Cells of interest, secs seconds, mins minutes. Source data are provided as a Source Data file.

We next tested our tissue structure and margin identification algorithms for region annotation in a dataset of pancreatic tissue samples from 12 patients. This dataset was profiled with IMC using a panel of 35 metal-tagged antibodies to investigate the spatial distribution of stromal and islet cell populations from individuals with long-term type 1 diabetes, those at the onset of the disease, as well as healthy control organ donors[13]. We used SPIAT's margin detection algorithm to identify islets and classify areas relative to the islets. With SPIAT, we identified a total of 1434 islets across samples (Fig. 6d and Supplementary Fig. 10), with an average of 1.87 islets per non-diabetic patient samples, 1.62 islets per disease onset sample, and 1.60 islets per long-term diabetes sample. Islet size decreased with disease progression (Jonckheere–Terpstra test (JT) one-sided $p$ value = 0.033) (Fig. 6e). The composition of islets also varied, where alpha, delta and gamma islet cell types increased in percentage as the disease progressed (one-sided JT $p$ value = $6.15 \times 10^{-3}$, 0.063, $9.93 \times 10^{-3}$, respectively), whereas beta cells decreased ($p$ value = $6.40 \times 10^{-4}$) (Fig. 6f), consistent with the findings of the original study[13] and the known biology and role of beta cells in diabetes[26]. Next, we investigated the spatial distribution of endothelial cells (EC), as it was not characterized in the original study. First, we classified regions as islet-infiltrated, internal margin, external margin, and stromal. EC tended to be increasingly located in the stroma as the disease progressed (one-sided JT $p$ value = 0.023), whereas EC in the infiltrated and external margin compartments decreased during progression (one-sided JT $p$ values = 0.015 and 0.03, respectively) (Fig. 6g). This trend was not observed in the internal margin (one-sided $p$ value = 0.11). We also observed a reduction in the average distance from infiltrated EC to the islet margin ($p$ value = 0.0020) and an increased distance between stromal EC and the islet margin ($p$ value = 0.14) (Fig. 6h), suggesting a process of migration of EC from the islets to the stroma. The crosstalk between EC and beta cells is well established, and dysfunction of EC has been associated with loss of beta cells during diabetes progression[27]. Our results suggest that the reduction of EC in islets and their enrichment in the stromal during diabetes progression could be an additional feature of EC dysfunction that accompanies beta cell loss in type 1 diabetes.

### Identification of cellular neighborhoods

The identification of "cellular neighborhoods", "communities", or "ecosystems" is gaining traction in the spatial analysis community, as these have been used to define novel disease subtypes[5]. Neighborhoods represent cells in an area with a specific spatial distribution, and often includes a mixture of cell types. Examples of cellular neighborhoods include the formation of cell aggregates or clusters with a specific composition, and cells with a dispersed spatial distribution[4,28,29].

SPIAT includes three algorithms for the detection of cellular neighborhoods, phenograph[30], dbscan[31], and a hierarchical-based algorithm to identify clustered and dispersed cells (Methods) (Supplementary Fig. 11). SPIAT also implements the average nearest neighbor index (ANNI), a statistical test to determine whether particular populations are significantly clustered or dispersed[32], which can be used to determine whether a neighborhood is present.

We analyzed a published cohort of 35 advanced-stage colon cancer tissue samples profiled with 56 markers using CODEX[28], each sample with images from four tissue regions, for a total of 140 images. This cohort included images from 17 patients with Crohn's-like reaction (CLR) pathology, characterized by the presence of tertiary lymphoid structures (TLS) (Fig. 7a—note large cluster with a concentration of B cells in the CLR example), and images from 18 diffuse inflammatory infiltration (DII) patients, without TLSs (Fig. 7a). We identified a total of 28 images with significant B cell clustering using the ANNI, consistent with the formation of TLS. CLR samples were significantly enriched in this group compared to DII samples, with 14/17 (82.35%) of CLR patients having at least one region with significant B cell clustering, compared to just 5/18 (27.78%) of DII patients (one-sided Fisher enrichment test $p$ = 0.0015), consistent with their known pathology (Fig. 7b). We next used SPIAT's hierarchical clustering algorithm to identify de novo clustered and dispersed cellular neighborhoods. As expected, we found that B cells were preferentially found in large clusters of more than 1000 cells in the CLR samples, but not in the DII samples (Fig. 7c).

We next investigated how the composition of cell types differed between the dispersed (Fig. 7d) and clustered populations (Fig. 7e). CD68+ macrophages, NK cells, and Tregs were more frequently found in the dispersed population (adjusted one-sided Wilcoxon test $p$ values = 0.0091, 0.020, and 0.020, respectively). Furthermore, the spatial location of immune subtypes was tied to the size of immune clusters in the neighborhoods. While CD8+ and CD45RO+CD4+ T cells were evenly distributed across clusters of all sizes (adjusted two-sided JT $p$ value = 0.10 for both) (Fig. 7f), other T cell subtypes, such as Tregs showed a decreasing trend (adjusted one-sided JT $p$ value = $2.96 \times 10^{-15}$) (Fig. 7g), suggesting diverse etiology of clusters linked to size.

Similarly, we also investigated the formation of clusters in the IMC diabetes[13] dataset. Two types of clusters were defined, composed of either immune cells or stromal cells, using the cell type definitions provided by the authors of the original study. The number of both stromal cell and immune cell clusters per image is highest at the onset of diabetes, compared to non-diabetic and long-duration patients (one-sided Wilcoxon rank-sum test $p$ value = 0.17, 0.073, 0.068, and 0.12, respectively) (Fig. 7h, i). However, EC were rarer in stromal clusters at the onset of the disease, compared to non-diabetic and long-duration diabetic patients (one-sided Wilcoxon rank-sum test $p$ value = 0.057 for both comparisons) (Fig. 7j). On the other hand, cytotoxic T cells (Tc) cells became rarer in immune clusters as the disease progressed (one-sided JT $p$ value = 0.13) (Fig. 7k).

These case studies showcase the power of SPIAT for a deep analysis of cellular neighborhoods, down to distinguishing the distribution of cell subtypes across neighborhoods and during disease progression.

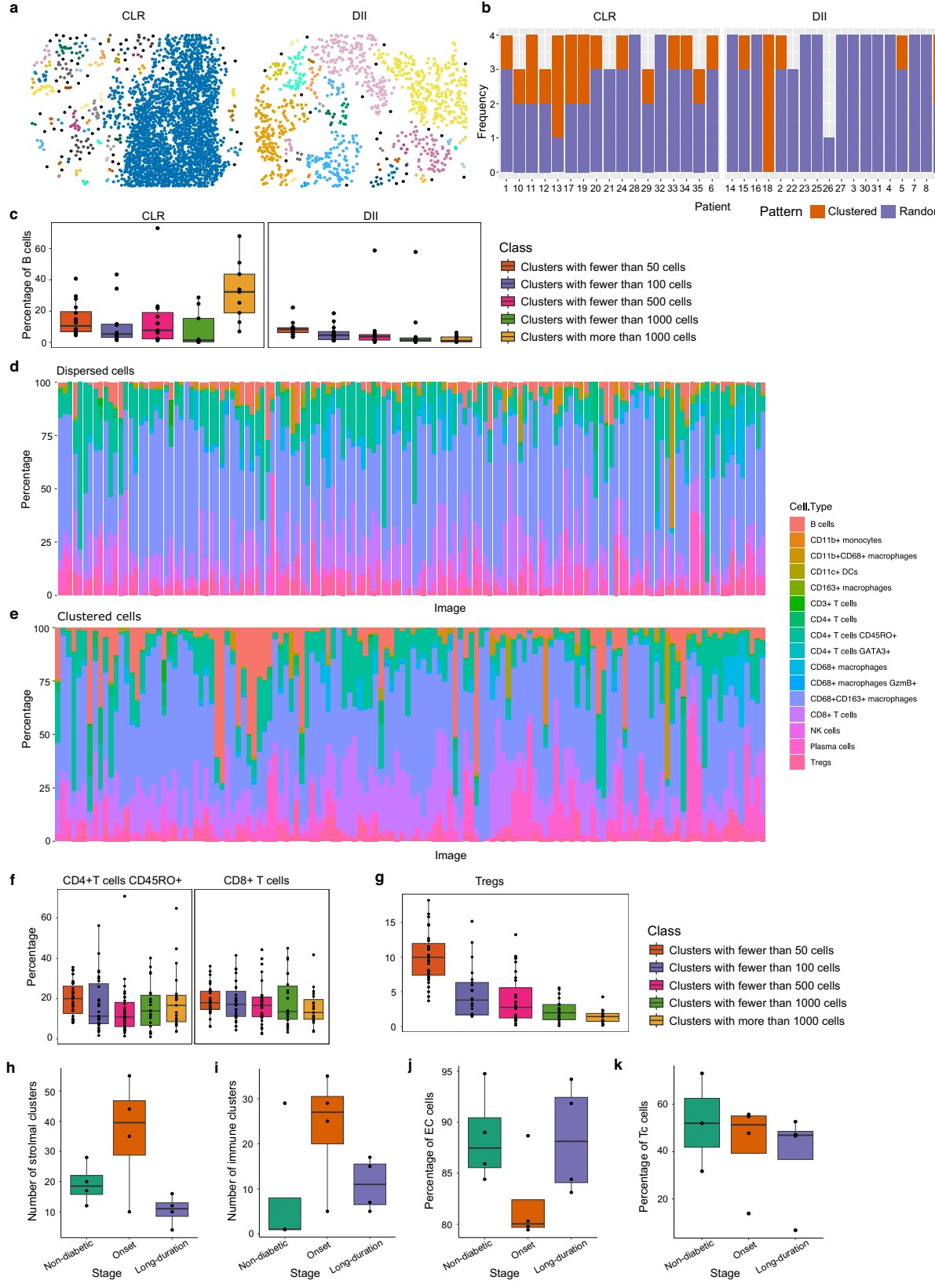

## Discussion

We present SPIAT, a platform-agnostic toolkit for the spatial analysis of tissues, and the first simulator of tissue spatial data, spaSim. With its six analysis modules, SPIAT includes common and novel algorithms for comprehensive spatial analysis within a single tool. The user-friendly interface of SPIAT simplifies installation and allows users to quickly perform comprehensive spatial analysis without needing significant computational power or learning advanced software frameworks. To our knowledge, spaSim is the first simulator of tissue spatial data, enabling, for the first time, to benchmark, compare and validate spatial

**Fig. 7 | Identification of cellular neighborhoods with SPIAT. a** CLR and DII example samples from the CODEX dataset. Cells are colored based on cluster membership. **b** Detection of significant clustering of B cells using the ANNI. Samples without B cells were excluded. **c** Higher percentage of B cells in the largest cluster in CLR samples, but not DII samples. $n = 126$ cell clusters computed from 35 biologically independent samples. **d** Percentage of immune cell types in the dispersed immune population. **e** Percentage of immune cell types in the clustered immune population. **f** The percentage of CD45RO$^+$ CD4$^+$ T cells and CD8$^+$ T cells across cluster sizes is similar. Adjusted two-sided JT $p$ value = 0.10 for both. $n = 268$ cell clusters computed from 35 biologically independent samples. **g** Larger clusters have lower percentages of Tregs. Adjusted one-sided JT $p$ value = $2.96 \times 10^{-15}$. $n = 120$ cell clusters computed from 35 biologically independent samples. **h** Average number of stromal clusters per sample by stage in the IMC dataset. Samples taken at the onset of diabetes have a higher number of stromal clusters (one-sided Wilcoxon rank-sum test $p$ values = 0.17 and 0.073, respectively).

**i** Average number of immune clusters per sample by stage. Samples taken at the onset of diabetes have a higher number of clusters of immune cells (one-sided Wilcoxon rank-sum test $p$ values = 0.068 and 0.12, respectively). **j** Percentage of endothelial cells in stromal clusters. The percentage decreases at the onset of diabetes, but returns to non-diabetic levels in the long-term diabetes samples (one-sided Wilcoxon rank-sum test $p$ value = 0.057 for both comparisons). **k** Percentage of Tc cells in immune clusters. The percentage of Tc cells decreases during progression (one-sided JT $p$ value = 0.13). **h**–**k** $n = 12$ biologically independent samples. In boxplots, the center line corresponds to the median and the box limits correspond to the first and third quartiles (the 25th and 75th percentiles). The upper and lower whiskers extend to the maximum or minimum value within 1.5 times the interquartile range, respectively. CLR Crohn's-like reaction pathology, DII diffuse inflammatory infiltration pathology, EC endothelial cells, Tc cytotoxic T cells. Source data are provided as a Source Data file.

metrics. Our methods require minimal user intervention, increasing speed and reproducibility in spatial analysis research. Methods in SPIAT and spaSim are not tissue-specific and we anticipate their applicability to include investigating and simulating solid tumors, normal tissues or tissues affected by other diseases, as well as tissues from a range of species. Our case studies in melanoma, prostate, colon, and breast cancer and diabetes showcase the power of SPIAT as a patient and cell neighborhood classification tool to identify clinically relevant known and novel subtypes across biological contexts.

To date, the lack of a tissue spatial data simulator has hampered method development and benchmarking of algorithms in the spatial analysis field. spaSim includes the building blocks needed to construct common spatial patterns, providing a flexible starting point from which users can generate additional patterns. From a few basic parameters, spaSim was able to simulate images that captured the key spatial relationships in a large diabetes dataset (Note N3), demonstrating its capability to reproduce the main spatial features of real tissue images. With spaSim, we compared popular and novel colocalization metrics based on their ability to capture clinically relevant patterns of the tumor microenvironment, revealing that no metric was a "one size-fits-all". A combination of complementary metrics and choosing the most appropriate algorithm based on patterns of interest is likely needed when assessing spatial interactions. We believe that spaSim and the benchmarking presented in this study will significantly facilitate this process.

SPIAT addresses many limitations in the application of spatial metrics as biomarkers. Our entropy gradients method allowed the self-contained classification of samples from a large TNBC dataset based on the attraction and repulsion of tumor and immune cells. The identified groups were significantly linked to prognosis, which was not reported in the original study. Entropy gradients could potentially allow prospective classification of samples in large tumor cohorts without relying on comparisons between samples or defining arbitrary thresholds.

Our intuitive metrics to measure spatial heterogeneity are a step forward from colocalization scores and allow the quantitative study of intra-tissue microenvironment heterogeneity. While cancer heterogeneity has largely been studied from the genomics perspective, studies of the tumor-immune microenvironment using spatial technologies have revealed a previously unappreciated heterogeneity in tumor-immune interactions. Our intuitive metrics to measure spatial heterogeneity allow characterization of the prevalence and distribution of spatial patterns within individual samples rather than average dominant patterns. Our Prevalence and Distinctiveness scores are complementary and allow dissecting the distribution of individual patterns, enabling the quantitative study of the heterogeneity of microenvironments.

Our automated detection of the tissue structures and margins, and classification of cells relative to these, allows independent profiling of each microenvironment sub-region. Importantly, these methods

require minimal user intervention, increasing robustness, speed, and reproducibility. Finally, our automated detection of cellular neighborhoods identified samples enriched in tertiary lymphoid structures and can identify both dispersed cellular neighborhoods as well as those with different levels of aggregation, which facilitates characterizing cellular ecosystems.

While we have presented benchmarking of methods and results with both simulated and real tissue data, we acknowledge that no method can be applied indiscriminately. Spatial analysis methods still require visual inspection of results to verify intuitive consistency with what can be observed. Similarly, we have included in spaSim the building blocks to construct the most common spatial patterns we have observed in in-house and publicly available data, providing a starting point from which users can generate additional patterns.

Development of SPIAT and spaSim will continue as spatial analysis further matures and further patterns of interest emerge, and may, in the future, support the analysis of 3D spatial data and incorporate machine learning and other strategies to further extend their capabilities. While, to date, the purpose of SPIAT and spaSim has been the deep characterization and simulation of spatial patterns in individual images, the development of methods to detect and compare spatial patterns across images will be the next frontier.

Overall, SPIAT and spaSim are powerful tools for the spatial analysis of tissue microenvironments that allows the identification and quantitation of spatial patterns associated with disease development and clinical outcomes, and the simulation of tissue spatial data to aid in method development. We anticipate SPIAT and spaSim will contribute to both the democratization of spatial analysis and the empowering of method developers, helping push forward the use of spatial metrics as biomarkers and to understand tissue biology.

## Methods
### Overview of SPIAT
The base object for SPIAT is SpatialExperiment[33], which was originally designed for spatially resolved transcriptomics data. Marker intensities are treated like gene expression levels, and cell coordinates and cell phenotypes (if available) as additional metadata. For ease of use, most functions are independent of each other and take the SpatialExperiment object as input. We have tested SPIAT successfully with images of up to ~1 million cells, and implementations of the algorithms were made to optimize speed.

SPIAT offers 6 analysis modules, including (1) Basic Analysis, (2) Visualization, (3) Cell Colocalization, (4) Localized metrics and spatial heterogeneity, (5) Neighborhoods and ecosystems, and (6) Tissue regions.

SPIAT was developed in R version 4.2 and is available at: [https://bioconductor.org/packages/SPIAT/] and [https://github.com/TrigosTeam/SPIAT]. The SPIAT tutorial is available at: [https://TrigosTeam.github.io/SPIAT/].

## Reading in data

The input to SPIAT is a table where rows are cell IDs, and columns include X, Y coordinates and marker intensities. Cell phenotypes can be optionally included if already available. Cells must have been segmented previously. SPIAT can take in the data generated by any software that generates the data required for input, such as, but not limited to, inForm, HALO, and CellProfiler. Users can input the data in a generic format of a data frame of coordinates and phenotypes (if available), and a data frame of marker intensities (if available). If cells were previously phenotyped, these can be used, but SPIAT also offers the option of de novo phenotype prediction based on marker intensity levels.

## Phenotyping of cells

We define "cell phenotype" as whether a cell is positive or negative for a marker (e.g., CD3$^+$ cells). Once "cell phenotypes" have been predicted for all cells, users can define "cell types" based on the combination of markers (e.g., CD3$^+$CD8$^+$ cells as cytotoxic T cells).

The de novo phenotyping of cells based on marker intensities does not require user intervention or the manual setting of thresholds. Our base algorithm assumes that most cells in an image are not positive for the marker of interest. With this assumption, we can estimate the background levels of the marker based on the distribution of marker intensities. Marker levels generally follow a distribution skewed to the left with a long right tail. The cutoff is selected as the inflection point of the distribution as it flattens.

In cases where the cells to be phenotyped are likely to represent most of the cells in the image, for example, tumor cells in a tumor section, and therefore their marker(s) are positive in a high proportion of cells (i.e., common marker), we have added an additional step to our base algorithm. Here, we first phenotype the rarer cell types based on the less common markers. This population of cells is used to determine the distribution of background levels of the marker of the more common population. Subsequently, we select the 0.95 quantile of the common marker in this rare population as a putative threshold (threshold 1). Next, we empirically determine the inflection point of the distribution of the marker of the common population using our base algorithm (threshold 2). Finally, we select whichever of the thresholds is greater as a cutoff for phenotyping.

## Visualization of tissues and basic analysis

SPIAT has multiple options for the visualization of the spatial distribution of cells, including visualization of cell types, cell phenotypes, and marker intensities. The latter allows the detection of staining artifacts as well as the intuitive detection of co-occurring or mutually exclusive markers.

SPIAT can calculate basic metrics, such as the proportion of cell types and distance-based metrics using Euclidean distances between cells. SPIAT compares the distribution of minimum distances between a pair of cell types, and calculates the mean, median and standard deviation. The same comparisons and summaries are also available for pairwise distances between all cells of two cell types. SPIAT also includes methods for the visualization of distances as heatmaps and violin plots.

SPIAT includes functions to plot interactive t-distributed stochastic neighbor embedding (tSNE) and Uniform Manifold Approximation and Projection (UMAP) plots based on marker intensities to verify cell type classification and identify and remove potentially misclassified cells.

## Cell colocalization

We define "cell colocalization" as the spatial relationship between two or more cell types. Cell colocalization defines whether cell types are distributed independently, co-occur in the same location or repel each other. SPIAT includes 7 colocalization metrics.

**Average pairwise distance (APD).** APD calculates the distance between all cells of one cell type against all cells of another cell type in an image. The distances between the two cell types of interest are subsequently averaged.

**Average minimum distance (AMD).** For each cell of the reference cell type, SPIAT identifies the distance to the closest cell of the target cell type. This process is repeated for each cell of the reference cell type. Distances are subsequently averaged.

**Cells in neighborhood (CIN).** We define the CIN as the percentage of a target cell type within a radius (i.e., neighborhood) of a reference cell type[25]. This calculation is carried out for each cell, and subsequently averaged. This method can be used to identify spatial structures by pinpointing cells with high neighborhood proportions of the target cell type.

**Mixing score (MS).** The mixing score was originally defined as the number of immune-tumor interactions divided by the number of immune-immune interactions[5]. We have generalized this score to allow calculation for any two cell phenotypes defined by the user.

$$\text{Mixing score} = \frac{n_{\text{ref to target interactions}}}{n_{\text{ref to ref interactions}}} \qquad (1)$$

Here, $n_{\text{ref to ref interactions}}$ is the number of interactions between reference cell types. $n_{\text{ref to target interactions}}$ is the number of interactions between reference cells and target cells. An interaction between two cells exists when the two cells are within a defined radius of one another. This radius needs to be specified by the user.

**Normalized mixing score (NMS).** The MS score is highly dependent on the total number of cells in the reference and target populations (Supplementary Fig. 2). To account for this, we added a scale factor to normalize it by the total possible interactions between cells in an image unlimited by radius.

$$\text{Normalized mixing score} = \frac{n_{\text{ref to target interactions}} * (n_{\text{ref}} - 1)}{2 * n_{\text{ref to ref interactions}} * n_{\text{target}}} \qquad (2)$$

$n_{ref}$ is the total number of reference cells, $n_{\text{target}}$ is the total number of target cells.

**Area under the curve of the cross K function (AUC).** Ripley's K function[34,35] is widely used to determine if the spatial distribution of point patterns across a study area is clustered, dispersed, or randomly distributed. It is defined as:

$$K(t) = \lambda^{-1}E \qquad (3)$$

Here, $\lambda$ is the intensity of the points, $E$ is the number of extra points within distance $t$ of a randomly chosen point. Then $K(t)$ is drawn as a curve against the increase of $t$. The shape of the curve shows the distribution pattern. Normally, a Poisson Process is used as a reference pattern to contrast the observed pattern. The cross K function (4) is a generalization of Ripley's K function for multi-type point patterns. It summarizes the spatial relationship between two or more types of points.

$$K_{(ij)}(t) = \lambda_j^{-1}E_{i,j} \qquad (4)$$

Here, $\lambda_j$ is the intensity of type $j$ points, $E_{i,j}$ is the number of type $j$ points within distance $t$ of a randomly chosen type $i$ point. If two point processes $i$ and $j$ are independent of each other, similar to a Poisson Process, then the expected cross K function should be $K_{(ij)}(t) = \pi t^2$. Comparing the observed cross K function with the expected cross K

function gives the spatial relationship between the two point processes. To quantify the patterns, we calculate the area between the curve of the cross K function and the expected Poisson Process $E$ by subtracting the area under the curves (AUC), resulting in a metric, $D_{AUC}$. $D_{AUC}$ is then normalized by the total area of the cross K plot. The normalization makes the comparison of this metric between images of different sizes possible. If $D_{AUC}$ is positive, the observed cross K curve is above the Poisson process curve and the two point processes are aggregated, indicative of high colocalization. If $D_{AUC}$ is negative, the observed cross K curve is below the Poisson process, indicating separation of the patterns and low colocalization. The calculation of the cross K function is carried out by the Kcross() function in R package spatstat[22].

**Cross K intersection (CKI).** This metric captures the aggregation of cells around a tissue structure, forming a ring-like pattern. For example, when there is an aggregation of immune cells in the tumor margin, forming an "immune ring" surrounding the tumor area. In these cases, the cross K function exhibits a crossing between the observed and expected curves. SPIAT defines this metric as the "Cross K intersection" (CKI).

$$CKI = 1 - \frac{x_1}{x_2} \qquad (5)$$

Here, $x_1$ is the x coordinate of the crossing in the cross K function graph; $x_2$ is the maximum x coordinate of the cross K function graph (Supplementary Fig. 12). The larger the crossing (larger $x_1$), the closer to the tissue structure area the crossing happens.

This metric is sensitive to the range of radii (and therefore $x_2$) considered in the cross K function. We suggest using one-quarter to half of the width of the image as the distance when defining cell rings. Extremely low levels of $x_1$ can be associated with patterns other than a cell ring. We treated cases where $x_1$ was lower than 0.04 as having no cell rings, based on our extensive testing. In these cases, the subtraction of $x_1$ and $x_2$ ratio from 1 results in low, but non-zero CKI scores. Even so, there are instances where there can be high CKI scores without the presence of cell rings (Supplementary Fig. 3).

## Localized entropy

Entropy measurements traditionally refer to the disorder or the diversity of entities in a system, in this case, cell types. However, since users set the markers and cell types to be profiled a priori, the number of cell types is fixed, and therefore entropy indicates the variability or imbalance in the numbers of cells in each cell type rather than diversity. Unlike traditional cell colocalization metrics that only involve two cell types, there is no limit to the number of cell types for the calculation of entropy, which therefore allows going beyond characterizing interactions between pairs of cell types.

The degree of disorder of one type of entity in the system is calculated by:

$$-\log_2 p(X_i) \qquad (6)$$

Here, $p(X_i)$ is the proportion of $X_i$ event in the whole system. If we are interested in several cell types, the weighted average of the entropies of interest indicates the overall disorder of all the cell types of interest.

$$Entropy = -\sum_{i=1}^{n} p(X_i)\log_2 p(X_i) \qquad (7)$$

We note that entropy itself does not provide spatial information. Rather, here we define a localized entropy to be used in combination with methods that define local regions, followed by calculating

patterns of the spatial distribution of localized entropy scores. Calculating localized entropies allows taking the spatial context into consideration (see Spatial Heterogeneity and Entropy Gradients sections below).

## Spatial heterogeneity

Tissue sections do not usually have one homogeneous pattern, but rather a mix of patterns in different areas of the image.

To measure this heterogeneity, SPIAT first splits the images into grids using a fishnet approach. Users can then choose the spatial pattern to be analyzed and the metric to detect it. In this work, we use localized entropy as it measures the balance in the number of cell populations of different types. The higher the balance, the higher colocalization in a small region. The following metrics can then be used to measure the prevalence of this pattern and its distribution in the image. Note that SPIAT also allows using other metrics besides localized entropy for characterizing different localized spatial patterns.

**Prevalence.** To quantify the Prevalence of a pattern in an image, SPIAT measures the percentage of the grid squares with the spatial pattern. For this, the user first selects a threshold for what constitutes the 'presence' of a pattern. For example, if we are using localized entropy, we might define the pattern as a ratio of 1:4 or more between two cell types in a grid square. A ratio of 1:4 between two cell types results in an entropy of around 0.72, so users can select 0.72 as the threshold. This is followed by the calculation of the percentage of grid squares, which denotes the prevalence of a pattern.

**Distinctiveness.** We define distinctiveness as the global spatial autocorrelation of a pattern. We calculate the global spatial autocorrelation of the squares using Global Moran's I[36]. The calculation is carried out by the moran() function in the R package elsa[37]. A high Distinctiveness score indicates the pattern is concentrated in particular regions of the tissue, whereas a low Distinctiveness indicates the pattern is dispersed throughout the tissue.

## Entropy gradient

We have introduced several colocalization methods that consider a circle of a fixed radius around reference cell types (CIN, MS, and NMS). However, a more robust approach is obtained by calculating a metric across a range of radii. The cross K function takes this approach and calculates the normalized total number of target cells within circles (number per unit area) based on an increasing set of concentric circles. Here we extend this concept further and propose the entropy gradient.

The entropy gradient function computes entropy as a function of radius and obtains the gradient for small radii. First, we select a reference cell type population and a range of radii. Starting from each reference cell, we identify the cells of the reference and target cell types within each of the concentric circles (defined by the range of radii). Next, we calculate the "aggregated entropy", which results in one entropy score for the entire image for each concentric circle radius. The aggregated entropy is calculated by using the total number of the reference cells and the total number of the target cells included in all circles of a radius as the quantities of the components in the entropy formula. Higher entropies are obtained when the cell types considered are balanced and suggest high cell colocalization, and lower entropies arise when particular cell type(s) are rare and indicate low colocalization.

Entropies are subsequently plotted against each of the radii. The slope of the resulting curve near radius = 0 indicates the spatial relationship between reference and target cell types. A negative slope indicates "Attraction" between cell types as the aggregated entropy is higher close to the reference population. In contrast, if the aggregated entropy increases as we go further from the reference populations, there will be a positive slope. We define this as "Repulsion". A slope of

zero can be obtained when a pattern is consistent across the image. We note that these patterns hold true when the reference population is a minority (smaller cell numbers) compared to the target cell population (see Fig. 4c, Note N4). We also note that, in theory, any colocalization score can be used with similar results (see Note N4). However, we propose using entropy as it allows us to characterize the colocalization of more than two cell types.

Since the definition of attraction and repulsion is based on the slope of the curve, the application of gradients allows an unbiased detection of repulsion and attraction of cell types enabling sample classification independent of other samples in the cohort.

### Defining regions relative to tissue structures

**Detection of the margin of a structure.** We define the margin of a structure as the cells in the periphery or border of a tissue structure. An example would be the margin of a tumor. Our margin detection algorithm is based on computing an alpha hull[38], a generalization of convex hull[39]. The convex hull of a set of points is a closed curve of a minimum perimeter that contains the whole point set (Supplementary Fig. 13).

Unlike convex hulls which use straight line segments to connect the outermost layer of points, alpha hulls use curves, and are therefore better suited for defining tissue structure margins (Supplementary Fig. 14). All the curves in the alpha hull are generated from circles of the same radius. The radius of the circles (α) decides the curvature of the alpha hull. A larger α gives a less curved alpha hull, and fewer points are recognized as bordering cells. SPIAT computes a default α value based on the number of reference cells in the image, but the user can also choose to specify the value of α. Larger values of α are better for tissue structures with very complex or unclear borders, whereas smaller values of α give a better resolution for those with clearer borders. We used the R package alphahull[40] to implement the alpha hull. The cells on the alpha hull are defined as the bordering cells of the structure.

Next, cells are assigned as either being "Inside", "Border", or "Outside" of the tissue structure by the R package sp[41,42] (Supplementary Fig. 15). First, our function draws a polygon that is composed of the bordering cells. Then, we use the point.in.polygon() function from the sp package to locate the cells. This function implements the ray crossing method to determine if a point is inside of a polygon or outside[43]. Starting from a point, a ray launches from the point. The number of intersections between the ray and the edges of the polygon indicates the location of the point. If the number is even, it means the point is outside of the polygon; odd, inside. As a result, each cell in the image is assigned an identity, "Inside", "Outside", or "Border".

We then identify the minimum distance of each cell to the margin, and further classify them as being in the core of the structure, outer or inner margin, or distal area. In the case of tumor structures, these would correspond to the infiltrating cells, external and internal margin, and stromal areas. Finally, SPIAT calculates the proportions of cell populations in these regions and summarizes distances from the cells to the margin.

**Ratio of border cell count to cluster cell count (R-BC).** The margin detection algorithm will run even in images without clear bordering cells. To determine whether the results of the margin detection algorithm are sensible, SPIAT calculates the ratio of border cell count to clustered cell count (R-BC) to identify images with tissue structures with a clear margin (Supplementary Fig. 16).

In cases of clear margins, the R-BC will be low as most of the cells of the tissue structure would have been categorized as "Inside" cells. On the other hand, the R-BC will be high in cases of poor or unclear margins as a relatively high number of cells will be miscategorized as bordering cells.

### Detection of cellular neighborhoods

SPIAT includes three algorithms for the detection of cellular neighborhoods, phenograph[30], dbscan[31], and a hierarchical-based algorithm for classification[29]. In the hierarchical-based algorithm, Euclidean distances between cells are calculated. This is transformed into a binary matrix of all cells against all cells, where 1 is given to pairs of cells that are within a specified threshold (interacting cells), and zero to the rest. This is then used to perform hierarchical clustering using hclust() with the single linkage method. Next, the dendrogram is cut using cutree() at the height of 0.5. The resulting branches constitute the clusters. We recommend choosing the threshold for interacting cells based on the average minimum distance between cells observed.

SPIAT allows the detection of clusters composed of specific types of cells defined by the user (e.g., immune cells) or clusters of cells regardless of cell types, commonly referred to as "communities", "neighborhoods", or "ecosystems". SPIAT returns the identity of each cell in the cluster and cluster size and can calculate cluster cell composition.

**Average nearest neighbor index (ANNI).** SPIAT includes a statistical test to test for the presence of clusters of cells, the average nearest neighbor index (ANNI)[32]. The ANNI evaluates the spatial aggregation or dispersion effect of objects based on the average distances between pairs of the nearest objects and can be used to test for the clustering of specific cell types (e.g., immune or tumor cells). Next, the $z$ score and $p$ value of the ANNI is calculated to validate the significance of the pattern.

The index calculates the ratio between the observed average distances ($D_o$) between pairs of nearest objects and the expected average distances ($D_e$) between pairs of nearest objects. $D_o$ is calculated from the real point pattern:

$$D_o = \frac{1}{n}\sum_{i=1}^{n} d_i \tag{8}$$

Here, $n$ is the total number of objects, $d_i$ is the distance between object $i$ and its nearest neighbor.

$D_e$ is calculated from a point pattern under complete spatial randomness (CSR) with the same intensity of the points with the observed point pattern. CSR is often modeled by a Poisson point process.

$$D_e = \frac{0.5}{\sqrt{n/A}} \tag{9}$$

Here, $A$ is the area of the 2D space. The ANNI is then calculated by:

$$\text{ANNI} = \frac{D_o}{D_e} \tag{10}$$

The index shows if the spatial distribution of one type of events is clustered (ANNI <1), random (ANNI ≈ 1) or dispersed (ANNI >1) (Supplementary Fig. 17).

A $z$ score is calculated to validate the significance of the pattern:

$$z = \frac{D_o - D_e}{\text{SE}} \tag{11}$$

SE is the standard error of the average distance between the expected pairs of nearest objects under CSR with the same intensity n/A. We used the constant number 0.26136 calculated by Clark and Evans (1954)[32] to compute SE.

$$\text{SE} = \frac{0.26136}{\sqrt{\frac{n^2}{A}}} \tag{12}$$

A $P$ value is calculated based on the $z$ score and indicates the significance of the pattern. We chose $5 \times 10^{-6}$ as the threshold to judge

significant patterns. Our package gives the final pattern based on the ANNI value and the *P* value. If the *P* value is smaller than $5 \times 10^{-6}$, an ANNI larger than 1 indicates a dispersed point distribution and an ANNI smaller than 1 indicates a clustered point distribution. If the *P* value is larger than $5 \times 10^{-6}$, there is insufficient evidence against random distribution. We note that the selection of the *P* value threshold can be arbitrary and suggest that the users adjust the threshold based on their images.

### Tissue spatial simulator (spaSim)

spaSim can generate highly customizable spatial patterns, including randomly distributed cells with specific proportions of cell types, tissue structures, clusters of cells, different levels of cell infiltration in tissue structures, aggregation of cells surrounding tissue structures, and external structures such as vessels. As each feature can be added separately, spaSim allows the generation of an infinite number of spatial pattern combinations. Images from spaSim can be directly used as input to SPIAT functions as they both use the SpatialExperiment as their base data structure. spaSim was developed in R version 4.2 and is available at: [https://bioconductor.org/packages/spaSim/] and [https://github.com/TrigosTeam/spaSim]. The spaSim tutorial is available at: [https://TrigosTeam.github.io/spaSim/].

### Simulating background cells

There are two spaSim models for simulating background population—the Hardcore process and the evenly spaced models.

Simulating background cells using a Hardcore process is obtained with the simulate_background_cells() function using the "Hardcore" method. Based on our experience, this model generates images that are comparable to tumor tissues since cancer cells have abnormal morphology and there is an overall loss of tissue structure. We use the rHardcore() function from the R package spatstat.random. A Hardcore process is simulated based on a Poisson process, with an additional step of eliminating points that are within a specified distance to any other point, as cells are distanced from each other based on cell volume. Our function uses an oversampling rate to create more cells than the target number of cells (specified by the parameter n_cells) to ensure the resulting image has the number of cells specified.

In contrast, the evenly spaced distribution model generated by simulate_background_cells() using the "Even" method assumes that cells are distributed approximately according to the vertices of a hexagon. We accomplish this by generating cells on a hexagonal grid and individually applying a bounded uniform random jitter. We have found that this model is more suitable when simulating normal tissues since normal cells are arranged in an organized, patterned manner within tissues.

### Simulating unstructured mixed cell populations

To create a random mixing of cell types, we use random number sampling to assign phenotypes to the background cells in specific proportions defined by the user. This can be done with the simulate_mixing() function.

### Simulating clusters

To define tissue structures and cell clusters, spaSim uses geometric shapes such as ovals, circles, and others to delineate regions of a specific cell type, where the user can customize the number of cells, the locations of the aggregates, and the combination of cell types present to simulate infiltration.

For tissue structures, such as tumor clusters, spaSim simulates ovals, circles, or combinations of ovals and circles. Given the center location and the radius (size) of the tissue structure, the simulator uses the mathematical formula for a circle/oval to find its margin. The cells inside the margin are defined as tissue-structure (e.g., tumor cluster) cells and the cells outside as non-tissue-structure cells (e.g., non-tumor

cells). To determine whether a cell is inside the shape, we calculate the distance from the cell to the center location of the shape. If the distance is larger than the radius (size) of the circle/oval, the cell is outside the cluster and should not be considered as a tissue structure cell. The number of cells can be customized by specifying the size of the cluster. If there are multiple cell types in the cluster (e.g. tumor cells and infiltrating cells), the assignment of identities to these cells is random, using the random number sampling technique based on proportions specified by the user.

To define an irregular shape, such as those of an immune cluster, we use part of a heart shape.

### Simulating cell rings

Cells sequestered in the periphery of the marker of the tissue structure, such as immune cells sequestered in the tumor margin ("immune rings"), are simulated using concentric circles and the difference between the radii (sizes) of the two shapes is the width of the ring. First, we specify the properties of the rings, such as their primary (inner cluster) and secondary (outer ring) cell types, size, shape, width, and location. Properties of cells infiltrating into the inner mass or outer ring can also be set. If there are multiple cell types lying in the cluster and the ring, the assignment of identities to these cells is random, using the random number sampling technique based on proportions specified by the user.

We can also simulate a double ring. Here, we aim to simulate a tissue structure with an inner ring (internal margin) and an outer ring (external margin). First, we specify the properties of double rings, such as their primary (tissue structure area), secondary (internal ring), and tertiary (external ring) cell types, size, shape, width, and location. Properties of cells infiltrating into the inner mass or either ring can also be set. If there are multiple cell types lying in the tissue structure cluster and the double rings, the assignment of identities to the cells is random, using the random number sampling technique.

### Simulating vessels

Here we aim to simulate stripes of cells representing blood/lymphatic vessels. These are simulated by using pairs of straight lines close to each other to define the walls of the vessel. First, we specify the properties of vessel structures, such as the number present, their width, and the properties of their infiltrating cells. We then randomly assign "background cells" which lie within these vessel structures to the specified cell identities in the specified proportions. The locations of the vessels are stochastic.

### Simulating a range of images varying by a specified parameter

In some cases, simulations of a set of images based on a range of values for a parameter are needed, especially when benchmarking. Rather than simulating images individually, simulating these images in one go is desirable. With spaSim we can simulate randomly distributed mixed cell types with different proportions of each cell type, multiple images with clusters of different properties (such as increasing size or increasing infiltration), and multiple images with cell rings of different properties, such as ring thickness.

To simulate images with clusters of different properties, users cannot manually define the base shape and the primary cell type of the clusters. Rather, we have predefined three options for the base shape available—the first two options are clusters of different shapes where the primary cell type is "Tumor" and there is infiltration of types "Immune" and "Others"; the third option includes an immune cluster in the stroma where the primary cell type is "Immune" and the infiltration cell types are "Immune1" and "Others". Users can then vary the parameters of each of these base shapes.

To simulate images with cell rings of different properties, there are also three predefined options for the base shape. The first two options are clusters with rings of different shapes where the primary

cluster cell type is "Tumor", the cluster infiltration cell types are "Immune" and "Others", the primary ring cell type is "Immune" and the ring infiltration type is "Others"; the third option is a cluster with a ring where the primary cluster cell type is "Tumor", the cluster infiltration cell types are "Immune" and "Others", the primary ring cell type is "Immune", and the ring infiltration type is "Tumor" and "Others". The cluster size, infiltration proportions, cluster location, ring width, and ring infiltration proportions can be defined in each.

## Datasets
We used four cancer datasets to showcase the capabilities of SPIAT.

**Prostate cancer dataset[29].** Twenty-six primary prostate cancer formalin-fixed paraffin-embedded (FFPE) tissue sections from radical prostatectomies from male patients diagnosed with prostate cancer were used. Samples were profiled using OPAL seven color multiplex immunohistochemistry (IHC), stained for DAPI, CD3, CD4, CD8, FoxP3, PDL1, and AMACR as a marker for prostate cancer cells. Images were scanned on the Vectra Polaris at 20X resolution. Multispectral image deconvolution, cell segmentation, and phenotyping were carried out with inForm Advanced Image Analysis Software (PerkinElmer, versions 2.3 and 2.4). For each tissue section, 14 to 16 representative regions of interest were selected, each with a window size of 2500 by 2000 pixels (1338 μm × 1004 μm). The table of cell coordinates, marker intensities, and phenotypes were exported from inForm, which was then used as input to SPIAT. The detection of the clustering of tumor cells was performed by R_BC() function. We ranked the images (with a minimum of 300 tumor cells and 300 immune cells) based on their R-BC scores. A low R-BC score indicates the presence of tumor clusters. Sample collection was approved by the Peter MacCallum Cancer Centre Human Research Ethics Committee. Informed consent was obtained from all participants. We complied with all ethical regulations.

**Melanoma dataset[44].** Samples were collected longitudinally from a female patient diagnosed with vaginal melanoma at diagnosis, at metastasis and at metastatic relapse. Samples were obtained during surgery and conserved as FFPE. Whole-tissue sections were profiled using OPAL seven color multiplex IHC, stained for T cells (CD3+), B cells (CD20+ and PDL1+/−), dendritic cells (CD11c+, PDL1+/−, and CD68+/−) and macrophages (CD68+ and PDL1+/−) and SOX10 as a marker for melanoma. Images were scanned on the Vectra Polaris at 20x. Multispectral images were deconvoluted with the inForm software (PerkinElmer) version 2.4.8. Multiple individual deconvoluted images were stitched together in the HALO image analysis platform (Indica Labs) version 3.0.311 with the HighPlex v2.0 module. Cell segmentation and phenotyping was also carried out with HALO. We exported the cell coordinates, phenotypes, and marker intensities from HALO as a table, which was then used as input to SPIAT. The automatic detection of tumor border was carried out by identify_bordering_cells() using tumor cells as the reference cell type and 100 as the threshold for cluster size, under which clusters were excluded. The identification of tumor structure was then performed with calculate_distance_to_margin() and define_structure() using T cells, B cells, dendritic cells, and macrophages as cell types of interest. The calculation of proportions of cell types in the tumor regions was performed by calculate_proportions_of_cells_in_structure(). The runtime of border detection and tissue structure identification of each melanoma image was recorded. Sample collection was approved by the Peter MacCallum Cancer Centre Human Research Ethics Committee. Informed consent was obtained from all participants. We complied with all ethical regulations.

**Triple-negative breast cancer (TNBC) dataset[5].** Breast cancer tissue was obtained from 39 female patients with triple-negative breast cancer. Thirty-six proteins in 40 FFPE tissue microarrays (TMA) (1 mm

cores) from biopsies were profiled with MIBI. Data in the form of a table of cell coordinates, marker intensities, and cell types were downloaded from [https://www.angelolab.com/mibi-data]. Cells were grouped based on major immune cell types. Markers of immune lineages, non-immune markers, and markers linked to immune checkpoint inhibitors were also excluded, as the purpose of the study was to investigate the interaction between major cell types. Only markers that were unique to a cell type, and not found in multiple cell types, were included. We used SPIAT's phenotyping tools and annotated CD3+ cells as T cells, CD4+CD3+, or CD4+ cells as helper T cells, CD8+CD3+, or CD8+ cells as cytotoxic T cells, CD68+ cells as macrophages, CD20+ cells as B cells, and MPO+ cells as neutrophils. Annotation of tumor cells was obtained from the annotations provided in the publicly available data. Microscopy images were obtained from: [https://mibi-share.ionpath.com/tracker/imageset].

To calculate the entropy gradients, we used the following values of distances: 50, 75, 100, 125, 150, 175, 200, 250, 300, 350, 400, 450, 500, 550, 600, and the entropy_gradient_aggregated() function in SPIAT. We then defined a positive or negative slope for each sample based on whether the entropy value at a distance of 50 was the highest of the series. Survival analysis was carried out using the survminer and survival R packages for Kaplan–Meier analysis. Patients with samples without the relevant cell types were excluded from the survival analysis. For the spatial heterogeneity analysis, we first used the grid_metrics() function in SPIAT with the cell types of interest being tumor cells and macrophages and the number of splits in the fishnet of each image being 20. We then used calculate_percentage_of_grids() to obtain the Prevalence score, using a threshold of 0.72, which corresponds to a ratio of 1:4 between two cell types (see "Prevalence" section under "Spatial Heterogeneity" above for further details), and calculate_spatial_autocorrelation() to measure the Distinctiveness score.

**Colon cancer dataset[28].** Thirty-five advanced-stage colorectal cancer (CRC) cases (17 with Crohn-like reaction pathology [ten females and seven males] and 18 diffuse inflammatory infiltrations [eight females and ten males]) were selected at random, matched for gender, age, and cancer type, location, and cancer stage. FFPE tumor sections were profiled with CODEX from 0.6 mm TMA cores, with four cores per patient. Fifty-six proteins were profiled in each TMA core. Data of cell coordinates and cell types were downloaded from [https://data.mendeley.com/datasets/mpjzbtfgfr/1]. The cell phenotypes available in the database were utilized. To investigate the presence of tertiary lymphoid structures (TLS), the ANNI was calculated for each of the four images per patient using B cells as the cells of interest. A one-sided Fisher's exact test was used for enrichment. To identify clusters, we used the identify_neighborhoods() function in SPIAT, using our hierarchical algorithm. The radius to determine if cells were proximal to each other was set at three times the average minimum distance between cells in the image. We defined clusters as containing cells of the following types: CD4+ T cells CD45RO+, CD68+CD163+ macrophages, plasma cells, CD8+ T cells, Tregs, CD4+ T cells, CD11c+ DCs, B cells, CD11b+CD68+ macrophages, NK cells, CD68+ macrophages GzmB+, CD68+ macrophages, CD11b+ monocytes, CD4+ T cells GATA3+, CD163+ macrophages, and CD3+ T cells. Only cells in clusters of at least ten cells were considered "clustered". The remainder of the cells were classified as "dispersed". The cell type composition was obtained by calculating the percentage of cells of each immune cell type in the dispersed and clustered populations of cells in each image. To compare the composition between the dispersed and clustered populations, we averaged the percentages across the four images per patient to obtain one value per patient for each cell types. One-sided Wilcoxon tests, followed by correction for multiple testing using the Benjamini & Hochberg method, was used to test for significant differences in the composition of the dispersed and clustered populations. Next, clusters were classified based on size: clusters with between 10 and 49 cells

were classified as having "fewer than 50 cells", between 50 and 99 as having "fewer than 100 cells", between 100 and 499 cells as having "fewer than 500 cells", between 500–999 as having "fewer than 1000 cells" and with 1000 cells and more as having "more than 1000 cells"[29]. Similar to the above, for each image, we calculated the composition of cells in clusters of each size category by calculating the percentage of cells of each cell type. We then calculated an average percentage across all images of a patient. Two- and one-sided Jonckheere–Terpstra tests (JT), followed by correction for multiple testing using the Benjamini & Hochberg method, were used to determine significant increasing or decreasing trends.

**Diabetes dataset**[13]. Sections from pancreatic tissue of four patients with recent onset type 1 diabetes (<0.5 years), four with long-standing type 1 diabetes (at least 8 years), and four non-diabetes control were used. Patients were matched by age and gender. Each group was composed of three male and one female patient. Cores were selected to include at least one islet. For each patient, between 26 and 45 cores were selected for each region of the pancreas, with two pancreas regions profiled for each patient. This resulted in 845 images in this dataset. Data acquisition was obtained with a Helios time-of-flight mass cytometer (CyTOF) coupled to a Hyperion Imaging System (Fluidigm). Samples were stained with 35 metal-tagged antibodies targeting key islet cell antigens and immune cell markers. We downloaded the table of cell coordinates, marker intensities, phenotypes, and images from [https://data.mendeley.com/datasets/cydmwsfztj/1] and [https://data.mendeley.com/datasets/cydmwsfztj/2].

For our analysis, we used the cell phenotypes and cell coordinate information provided by the original study. We used SPIAT's identify_bordering_cells() function with an alpha hull of 20 to identify the margins of the islets. We defined the reference cell types for the structure of interest, being the cells annotated as "islet" in the data. We next used the identify_bordering_cells() function to calculate the number of islets per image.

We next aimed to define tissue regions relative to islets. For this, we used the define_structure() function, where the thickness of the internal and external margins surrounding the bordering cells was set at five cells. With this, we obtained a classification of cells based on areas relative to islets ("inside", "border", and "outside", as well as a more refined classification of "stroma", "external margin", "internal margin", and "infiltrated").

To determine islet size, we added the number of alpha, beta, delta, and gamma cells that were classified as being "inside" or "border" as these would be part of the islet structures, and then averaged across all images of each patient. To measure how the proportions of these cells changed as the disease progressed, we calculated the proportion of each of these cell types in the islets of each image, and averaged per patient.

We next calculated how endothelial cells were distributed relative to the structure by calculating the percentage of endothelial cells in each region relative to the total number of endothelial cells in an image using calculate_proportions_of_cells_in_structure() and selecting the "The_same_cell_type_in_the_whole_image" output. Percentages were averaged across images of individual patients and then compared based on disease stage.

To calculate the distance of endothelial cells to the tumor margin, we selected images with at least ten endothelial cells. We used SPIAT's calculate_summary_distances_of_cells_to_borders() to calculate the mean distance of cell types to the margin. This distance is calculated separately for cells outside of the tissue structure (stroma) and for cells within the tissue structure (in this case islets). Next, we averaged the distances of endothelial cells in the stroma and in islets to the margin of all images of a patient.

In all cases, one-sided JT tests were used to determine the significance of the trend during disease progression.

## Statistics

One-sided JT tests were used to test for increasing or decreasing trends on categorical data. Two-sided JT tests were used to check for trends in either direction. One-sided Fisher enrichment tests were used to test for enrichment. Correction for multiple testing was done using the Benjamini–Hochberg procedure. One and two-sided Wilcoxon rank-sum tests were performed to test the statistical significance between two samples of data. Kaplan–Meier analysis was used to measure the association between patient classification based on entropy gradients and time to death. Boxplots were generated with geom_boxplot() from the ggplot2 R package using the default values to determine the center line, box limits, quartiles, whiskers, and outlier points. Linear regression was performed on the comparison between metrics of real and simulated diabetes images, and adjusted $R^2$ was used as a proxy of the strength of the similarity between real and simulated datasets.

## Reporting summary

Further information on research design is available in the Nature Portfolio Reporting Summary linked to this article.

## Data availability

The MIBI TNBC data used in this study are available at [https://www.angelolab.com/mibi-data]. The CODEX colon cancer data used in this study are available at [https://data.mendeley.com/datasets/mpjzbtfgfr/1]. The IMC diabetes data used in this study are available at [https://data.mendeley.com/datasets/cydmwsfztj/1] and [https://data.mendeley.com/datasets/cydmwsfztj/2]. The prostate cancer and melanoma data are available under restricted access as our patient consent does not allow depositing data online, access can be obtained by contacting the corresponding author. Source data are provided with this paper.

## Code availability

The SPIAT package is available at [https://github.com/TrigosTeam/SPIAT] as well as in Bioconductor ([https://bioconductor.org/packages/SPIAT/]). The SPIAT tutorial is available at: [https://trigosteam.github.io/SPIAT/]. The spaSim package is available at [https://github.com/TrigosTeam/spaSim] as well as in Bioconductor ([https://bioconductor.org/packages/spaSim/]). The spaSim tutorial is available at: [https://trigosteam.github.io/spaSim/]. The data analysis and simulation code to reproduce all results of the manuscript is available at: [https://github.com/TrigosTeam/SPIATspaSimNCCodeShare][45].

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

## Acknowledgements

This work was supported by National Health and Medical Research Council Ideas Grants 2003887 and 2003115 awarded to A.S.T. We thank Prof. Alicia Oshlack for critical reading of the manuscript, Mr. Richard Young and Dr. George Au-Yeung for initial discussions at the start of this project, Mr. Ryan Sam for providing inspiration for spaSim, and Ms. Wei Ni Lim and Thu Ngoc Nguyen. A.S.T. is a Prostate Cancer Foundation Young Investigator. We acknowledge the Centre for Advanced Histology and Microscopy (CAHM) at the Peter MacCallum Cancer Centre for their support of this work. We thank the Research Computing Facility at the Peter MacCallum Cancer Centre for providing the infrastructure and support to carry out this project.

## Author contributions

Y.F. developed algorithms and software, analyzed the data and interpreted results, wrote the manuscript, and prepared figures, T.Y. developed algorithms and software, J.Z. developed algorithms and software, M.L. developed algorithms and software, M.D. developed software, V.O. developed algorithms and software, G.B. developed algorithms, A.Pi. generated data, L.C. developed software, S.W. developed software, A.Pa. generated data, N.K. generated data, Y.K.H. developed algorithms, S.P.K. designed the study, T.P.S. designed the study, P.J.N. designed the study, R.B.P. designed the study, S.S. designed the study, D.L.G. designed the study, A.S.T. conceived and designed the study, acquired funding, supervised the work, analyzed the data, interpreted results, and wrote the manuscript.

## Competing interests
The authors declare no competing interests.
