## [Peer Review File · Nature Communications]

Spatial analysis with SPIAT and spaSim to characterize and simulate tissue microenvironmentsREVIEWER COMMENTS

Reviewer #1 (Immune cell biology, spatial imaging method) (Remarks to the Author):

This paper presents a new R package for the spatial analysis of cells in the tumor microenvironment. The package is designed for the analysis of data generated with spatial proteomics technologies such as OPAL, CODEX and MIBI.

I do not have extensive expertise in software implementation, so my review mainly focuses on the value and impact of the software. Overall the package fill an unmet need in the unbiased analysis of cell location within tissues. I actually think there is a missed opportunity by restricting their narrative and testing to the tumor microenvironment. It seems the package could be used more broadly, and I would encourage the authors to consider demonstrating the use of their package with other types of data, or at least indicating the potential of their package in other conditions.

When I tried to test the software, it took me a while to find the last version (SPIAT 0.99.1). The link provided in the manuscript instructs to install SPIAT with Bioconductor, which does not seem to have SPIAT (even in devel version). Google search led me to former versions of the package, including a vignette for v0.4 which it does not exist? I finally found it on rdrv.io. It is important to tidy this up, especially because this package could be of interest to scientists that do not necessarily have lots of coding experience. Similarly, reading in data could be easier, if there was a way to upload an excel or cvs file. Imaging softwares often let you download cell position and marker intensities in those types of format. Having this option (with a table example to help the user format the file) would again help users with little R experience.

I found the paper and vignettes clear. One think that was not clear to me was whether you could input the cell types and location only, in case the analysis was done in another software, and still analyse the cell location (maybe that could be a vignette). I was also wondering whether the software could handle 3D data? And if not, whether there is a specific reason of for why it could not. In Figure 3, the CKI method looks promising to quantify exclusion. Is there any other situations for which the CKI would also give a high value? That's important to understand the limits or types of location patterns that can be revealed by this value. In Figure 4 and 5, to which extend is the entropy value dependent on cell density? If there are 50% less T cells but localisation is similar, how would the entropy values differ?

Reviewer #2 (Multiplex image analyses, cancer therapy) (Remarks to the Author):

The authors are commended for developing free R packages for spatial analysis.

The authors should note that Figure 1A refers to staining methods. What your R package requires is the output of the quantitative imaging platform (Figure 1C) that is used to extract data from those staining methods that you mention in Figure 1A. Suggest rewording Figure 1A figure legend to more accurately and better reflect this point.

SPIAT Tool

The SPIAT tools that are created are present in expensive commercial spatial analysis software such InFORM, HALO, StrataQuest, Definiens Tissue Studio (before they were bought by AZ), etc. However, having a validated R package with these features instead would be useful for the field.

The manuscript is basically a description of the tools which many in the field are currently creating including the commercial groups mentioned above. What would be more useful for your audience is a use case with 100s to 1000s of training images and a validation set with 100s to 1000s of images. For the SPIAT tool the small number of cases (40 – 56) is insufficient to test the ability of your tool to deal with the heterogeneity of the tumor immune microenvironment for each feature.

The digitized images in all of your figures are not convincing without the actual original stained image for comparison. In Figures 4F, 6G, and 6H what type of survival is being measured.

spaSIM Tool

Simulated tools need to be verified and validated. It is unclear what ground truth was used. As above, there should be a viable training set and a validation set. There are a lot of unanswered questions such as what percentage of the simulated features will be artefacts of the simulation process versus real features that happen in real life. This is a common question for simulated tools.

Materials & Methods

This manuscript is lacking a dedicated materials & method section. The patient populations and samples used need to be better described. For example, it is unclear how many fields of view (FOVs) were in each case and how many of those FOVs were included in the data analysis. Are these full biopsies, resections, TMAs? What were the inclusion and exclusion criteria for the phenotypes and markers? What statistics are used? Both tools are obviously running some level ML and other computational methods, but nothing is described about these methods. Etc., We are missing a lot of information here.

Discussion

The discussion in general was superficial. There are a lot of limitations to both of these tools that need to be discussed in detail.

**Rebuttal for "Spatial analysis with SPIAT and spaSim to characterize and simulate the tumor immune microenvironment" (Manuscript ID NCOMMS-22-17699-T)
(renamed to "Spatial analysis with SPIAT and spaSim to characterize and simulate tissue microenvironments")**

Reviewer #1:

Reviewer comment	Response	Location of update
This paper presents a new R package for the spatial analysis of cells in the tumor microenvironment. The package is designed for the analysis of data generated with spatial proteomics technologies such as OPAL, CODEX and MIBI. I do not have extensive expertise in software implementation, so my review mainly focuses on the value and impact of the software. Overall the package fill an unmet need in the unbiased analysis of cell location within tissues.	We thank the reviewer for appreciating that our packages address an important current gap in the field.	N/A
I actually think there is a missed opportunity by restricting their narrative and testing to the tumor microenvironment. It seems the package could be used more broadly, and I would encourage the authors to consider demonstrating the use of their package with other types of data, or at least indicating the potential of their package in other conditions.	We thank the reviewer for this suggestion. We have generalised our tools and the manuscript to the analysis of tissue sections and now present the analysis of the tumor microenvironment as an example application. Furthermore, we have added an additional case study in our results section investigating the pancreatic immune and stromal microenvironment in diabetes (lines 315-339, Figure 6d-h, lines 352-359, and lines 395-401, Figure7h-k, lines 418-423) and have reframed our colon cancer case study to focus on the distribution of immune populations (lines 373-393, Figure 7a-g, lines 407-418). These case studies now showcase how SPIAT can be used to identify changes in the microenvironment during the development and progression of diabetes, as well as to characterise the spatial distribution of immune and stromal populations. We have also added additional capability to spaSim to allow the simulation of cells from normal tissues, by including a novel simulation procedure	We have edited the manuscript throughout to reflect this change, as well as updated the packages and tutorials to use more generic terms, such as green and red, instead of tumour and immune, and tissue structure instead of tumor area. Substantial changes made: Title (lines 1-2) Abstract (lines 21-22, 28-29) Introduction (lines 46-53, 73-78) Results (lines 103-107,

	to simulate the locations of normal cells (Methods: lines 852-870). We also included extensive validation of our simulator using the diabetes dataset, demonstrating its capability to capture and simulate patterns beyond that of the tumor microenvironment (Supplementary Note N3). We updated our packages and replaced specific reference to tumor or immune cells in the functions to more generic terms. The tutorials (https://trigosteam.github.io/SPIAT/articles/SPIAT.html) has also been updated to reflect these changes. Finally, the Discussion now includes a discussion of this point.	138-143, 189-193, 315-339, 352-359,373-393, 395-401, 407-423) Discussion (433-438, 440-445, 490-492) Methods (lines 852-870) Added new Supplementary Note N3
When I tried to test the software, it took me a while to find the last version (SPIAT 0.99.1). The link provided in the manuscript instructs to install SPIAT with Bioconductor, which does not seem to have SPIAT (even in devel version). Google search led me to former versions of the package, including a vignette for v0.4 which it does not exist? I finally found it on rdrv.io. It is important to tidy this up, especially because this package could be of interest to scientists that do not necessarily have lots of coding experience.	We apologize for the difficulty the reviewer had in identifying the current version of our tools. The SPIAT and spaSim have now been accepted by Bioconductor, and are now available on Bioconductor's website: https://bioconductor.org/packages/release/bioc/html/SPIAT.html https://bioconductor.org/packages/release/bioc/html/spaSim.html These are the instructions for downloading: <pre>if (!require("BiocManager", quietly = TRUE)) install.packages("BiocManager") BiocManager::install("SPIAT") BiocManager::install("spaSim")</pre> The tutorial of SPIAT can be found here: https://trigosteam.github.io/SPIAT/articles/SPIAT.html The tutorial of spaSim can be found here: https://trigosteam.github.io/spaSim/index.html We have also made available all the code used in this manuscript here: https://github.com/TrigosTeam/SPIATspaSimNCCoDeShare	We have updated the landing page of our Github to make it easier for users to find and download the latest versions of the packages.
Similarly, reading in data could be easier, if there was a way to upload an excel or cvs file. Imaging softwares often let you download cell position and marker intensities in those	SPIAT can read in data formatted as a table from any format that is readable by R (including csv, txt and excel). An example of how to read this type of data into SPIAT is included in our online tutorial under 'Reading in data through the 'general' option' (https://trigosteam.github.io/SPIAT/articles/data_r	Results (lines 86-88, 116-124)

types of format. Having this option (with a table example to help the user format the file) would again help users with little R experience.	eading-formatting.html#reading-in-data-through-the-general-option-recommended). There are also options of reading in data processed with InForm and HALO under 'Reading in data pre-formatted by other software' in the tutorial. We have also made this point clearer in the manuscript.	
I found the paper and vignettes clear. One thing that was not clear to me was whether you could input the cell types and location only, in case the analysis was done in another software, and still analyse the cell location (maybe that could be a vignette).	Thank you for pointing this out. Yes, users can read in data with just the cell type and cell locations using the 'general' format. An example of how to read this type of data into SPIAT is included in our online tutorial under 'Reading in data through the 'general' option'. We have also better explained this in the results section of the main manuscript.	Results (lines 116-124)
I was also wondering whether the software could handle 3D data? And if not, whether there is a specific reason of for why it could not.	SPIAT currently cannot analyse 3D data. Our functions are designed to handle two axes, and the analysis of 3D data will require all our functions to handle an additional axis, as well as, importantly, re-imagining many of the algorithms and developing others to take advantage of the information provided by a third dimension. Unfortunately, to our knowledge, there are no high-throughput, large-scale 3D spatial proteomics or transcriptomics technologies commercially available on the market. As a result, datasets of multiplex spatial 3D data are scarce and are generally generated from serial sections of tissue, resulting in just a few coordinates (often < 5) in the additional dimension, whereas the X and Y coordinates often have thousands of data points. This unevenness in data richness across the axes severely limits the power of a 3D spatial analysis tool. We agree with the reviewer that this is the next frontier for spatial analysis, with technologies likely moving in this direction in the coming years. We have plans to develop a version of SPIAT and spaSim for the analysis and simulation of 3D data in the near future, and currently have grant applications under consideration to enable this work. We have added this commentary to our discussion.	Discussion (lines 483-485).

	Even so, we believe that the novel analysis tool and simulator for 2D data described here will greatly facilitate analysis and interpretation in the plethora of studies now using and reporting 2D analysis of spatial interactions.	
In Figure 3, the CKI method looks promising to quantify exclusion. Is there any other situations for which the CKI would also give a high value? That's important to understand the limits or types of location patterns that can be revealed by this value.	We thank the reviewer for this suggestion. To better understand the behaviour of the CKI method, we tested the CKI method across a range of simulations to investigate cases where the CKI gives high values. We now include more extensive characterization of this metric, and report the cases and incidences where a high CKI value might be obtained across a range of spatial patterns. This can be found in Figure S3. Briefly, we found that when there are target cells (e.g. immune cells) surrounding a defined structure (e.g. tumor cluster), high values are obtained over 99% of the time, which is what is expected of the CKI. However, we do note some scenarios where CKI values can appear higher than intuitively expected. For example, high CKI values can occur in the absence of defined reference structures (e.g. tumor clusters), or where there are few immune cells surrounding the tumour clusters. However, even in these scenarios, immune cells are present near reference/tumour cells, which is what the CKI metric captures. This is consistent with our message that spatial metrics cannot be used blindly or in isolation, and the combination of metrics is key to obtain an accurate representation of pattern (Results: lines 180-182, Discussion: lines 476-478). For example, scores obtained with the CKI could be interpreted together with metrics to measure the levels of clustering of tumor cells, and the amount of immune cells in the margins of the tumor to further validate the presence of immune exclusion.	We have added a new supplementary figure: Figure S3. We also added text in the results section, lines 176-178.
In Figure 4 and 5, to which extend is the entropy value dependent on cell density? If there are 50% less T cells but localisation is similar, how would the entropy values differ?	In general, the calculation of entropy depends on the relative proportion of one cell type to other cell types of interest in an area, rather than the exact proportion of an individual cell type. If the proportions of all the cell types in an area are the same, then the entropy is at its maximum. If one of the populations decreases its proportion, becoming rarer than the others, the entropy decreases. Similarly, if one of the populations	We have added a new supplementary figure: Figure S5. Results (lines 196-201) See Note N4.

	increases its density, becoming more common than the others, the entropy also decreases. Hence, entropy measures how distinct the proportions of different cell types are in an area. We have added a new Supplementary Figure (Figure S5) where we show simulations demonstrating these scenarios, and have added this explanation to lines 196-201 in the results. In our paper we apply the value of the entropy to two settings, the calculation of entropy gradients (Figure 4) and when calculating spatial heterogeneity (Figure 5). In Figure 5 we utilize grid metrics with entropy, and therefore the trends described above will be observed within each grid even if the location of the cells remains the same. Supplementary Figure S5 shows simulations demonstrating these scenarios. In Figure 4 we utilize the gradient with aggregated entropy. While the absolute value of the aggregated entropy will vary as discussed above, given that the results of attraction and repulsion of cell types depend on their location, rather than density, the levels of repulsion and attraction between cell types does not change. This demonstrates the robustness of this metric. Characterization of the effect of the proportion of a cell type on the gradient with aggregated entropy is discussed in Note N4.	
--	--	--

Reviewer #2:

Reviewer comment	Response	Location of update
The authors are commended for developing free R packages for spatial analysis.	We thank the reviewer for their support.	N/A
The authors should note that Figure 1A refers to staining methods. What your R package requires is the output of the quantitative imaging platform (Figure 1C) that is used to extract data from	We thank the reviewer for bringing this to our attention. We have updated the legend of Figure 1 to better explain that panels A, B and C correspond to data generation and processing done before input to SPIAT, whereas panel D indicates the start of SPIAT's functionalities.	Lines 116-130

those staining methods that you mention in Figure 1A. Suggest rewording Figure 1A figure legend to more accurately and better reflect this point.		
The SPIAT tools that are created are present in expensive commercial spatial analysis software such InFORM, HALO, StrataQuest, Definiens Tissue Studio (before they were bought by AZ), etc. However, having a validated R package with these features instead would be useful for the field.	We thank the reviewer for appreciating that our packages address an important current gap in the field, and that our package will help in the democratization of these tools, which to date have been only commercially available.	N/A
The manuscript is basically a description of the tools which many in the field are currently creating including the commercial groups mentioned above. What would be more useful for your audience is a use case with 100s to 1000s of training images and a validation set with 100s to 1000s of images. For the SPIAT tool the small number of cases (40 – 56) is insufficient to test the ability of your tool to deal with the heterogeneity of the tumor immune microenvironment for each feature.	We would like to clarify that SPIAT does not use machine learning, nor does it estimate parameters to fit a model, and therefore, the concepts of training and validation sets are not directly applicable. Since there is no need to train algorithms or fit models, for which sample sizes in the order of the 1000s would indeed be required, SPIAT can handle the relatively small sample sizes of the case studies presented. Methods in SPIAT are based on deterministic algorithms, spatial statistics, and mathematical equations that derive metrics of particular spatial features from individual images. We have added this clarification of the basis of SPIAT in the main results section, when SPIAT is first introduced (lines 88-89), and full details of the methods can be found in the Materials and Methods section, lines 498-843. Most projects using spatial multiplex technologies (MIBI, CODEX, image-mass cytometry, multiplex immunohistochemistry) have sample sizes in the order of 10-60 samples due to costs, challenges in sample collection, etc. Therefore, there is great value in showcasing the power of our tool in sample sizes representative of those of the vast majority of users.	We have included in the Results a description of the basis behind methods in SPIAT (lines 88-89) Refer to the Materials and Methods section for a full description on the methods used in SPIAT (lines 498-843) Refer to lines 250-280 for SPIAT’s Spatial Heterogeneity Analysis Module.

	We consistently show through our case studies that SPIAT can derive expected and novel spatial patterns linked with disease progression and prognosis, with broad applications across biological contexts. SPIAT is also capable of accommodating heterogeneity from multi-region sampling (colon cancer and diabetes case studies, Figures 6 and 7). This indicates that SPIAT is able to find meaningful biological signal within the vast heterogeneity of tissue microenvironments. Given that heterogeneity of the microenvironment can also occur within tissues, SPIAT also includes an analysis module to measure spatial heterogeneity within individual images, which allows measuring how specific spatial patterns can be unevenly distributed (see lines 250-280).	
The digitized images in all of your figures are not convincing without the actual original stained image for comparison.	We have included in the supplementary material the original microscopy images showing the single staining and the resulting composite image of our in-house datasets, namely, the prostate cancer and melanoma images (digitized images shown in Figures 2b, 6b, 6c, microscopy images shown in Figure S1, S7, S8 and S9). For the TNBC MIBI dataset by Keren et al, the corresponding microscopy images for all the computer-rendered images shown in our manuscript (Figures 4d and 5c) were obtained from https://mibi-share.ionpath.com/tracker/imageset and have been included in the supplementary material (Figure S4). For the colon cancer CODEX dataset by Schurch et al., the original microscopy images are unfortunately not publicly available in a format that can be displayed by us. We have used the cell coordinates and cell phenotypes made available from the original study, which has already been peer-reviewed (https://doi.org/10.1016/j.cell.2020.07.005). For the diabetes IMC dataset by Damond et al., the single-channel microscopy images of the computer-rendered image shown in Figure 6d can be found in Figure S10. These were downloaded from https://data.mendeley.com/datasets/cydmwfsztj/2.	We have added five new supplementary figures: Figures S1, S4, S7, S8, S9 and S10

In Figures 4F, 6G, and 6H what type of survival is being measured.	Figure 4f corresponds to a survival analysis of the time to death. Upon updating the colon cancer case study following suggestions from Reviewer #1, we have now excluded Figures 6G and 6H from the manuscript.	Clarified the type of survival being measured in lines 227-230 and 247
Simulated tools need to be verified and validated. It is unclear what ground truth was used. As above, there should be a viable training set and a validation set. There are a lot of unanswered questions such as what percentage of the simulated features will be artefacts of the simulation process versus real features that happen in real life. This is a common question for simulated tools.	We agree that quantitative validation of our spaSim simulator is important and have added this to our manuscript. We note, however, that training and validation sets per se are not applicable in our case as spaSim does not use machine learning. The purpose of spaSim is the testing of spatial metrics in a clean and controlled environment to understand the behaviour of metrics across different ranges of spatial patterns generated with different parameter settings, as we have done in Figures 3-5. The spatial features that spaSim aims to capture are the co-localization of cells, the formation of clusters and cell proportions. Simulated images are constructed algorithmically based on basic input parameters, such as number of clusters and cell proportions. We have added additional description of spaSim and its intended use in lines 101-103 and 110-113 of the main manuscript. Therefore, to validate spaSim, we have focused on its ability to generate simulated images with cell proportions, and co-localization and clustering metrics similar to those from a reference dataset of real images. Our results show that the features derived from the simulated images are quite comparable to those of the real images, indicating that the images simulated by spaSim indeed capture the spatial patterns of interest to us that are found in real images. Furthermore, we include quantification of the similarity between features derived from real and paired simulated images using R^2 values so that users and readers have an understanding of the strengths and limitations of spaSim. A detailed description of our validation can be found in a new supplementary note, Supplementary Note N3.	We have added a new supplementary note for the validation of the simulation tool: Note N3. We have updated lines 101-103, 110-113 in the results to include a more comprehensive description of the purpose of spaSim.

This manuscript is lacking a dedicated materials & method section.	We have added the Materials and Method section to the Main Manuscript, after the Discussion section, as requested by the formatting instructions of the journal. Here we describe the algorithms, mathematical formulas, and use cases of all functions in SPIAT. We also include details on the data analysis steps used for the case studies.	Added Materials and Methods section to the main manuscript file (lines 496-1083)
The patient populations and samples used need to be better described. For example, it is unclear how many fields of view (FOVs) were in each case and how many of those FOVs were included in the data analysis. Are these full biopsies, resections, TMAs? What were the inclusion and exclusion criteria for the phenotypes and markers? What statistics are used?	We have included the description of each of the datasets in the Materials and Methods, including the source of the tissue, the number of samples and fields of view. Furthermore, we have also included how we handled cases where there were multiple fields of view for a single patient and related details, and the criteria used for the inclusion of phenotypes and markers in our analyses. The specific statistics used are also included and are also pointed out throughout the results section.	Lines 945-1083
Both tools are obviously running some level ML and other computational methods, but nothing is described about these methods. Etc., We are missing a lot of information here.	Neither SPIAT nor spaSim is using machine learning. The methods are based on deterministic algorithms, spatial statistics, and mathematical equations. The full details of the methods can be found in the Materials and Methods section, and we have added clarification of this when the tools are first introduced in the results section (lines 88-89, 101-103 and 110-113).	Materials and Methods section (lines 496-1083). Results (lines 88-89, 101-103 and 110-113)
The discussion in general was superficial. There are a lot of limitations to both of these tools that need to be discussed in detail.	We have extended our discussion and provide further details of the context in which the tools can be used, as well as future work.	Lines 426-495

Additional updates

Upon review of our packages by Bioconductor, it was suggested to us that SPIAT and spaSim should use the SpatialExperiment object, rather than the SingleCellExperiment object. We have therefore updated both packages accordingly. This was a software engineering change, and does not change the algorithms, methods, or features presented in this work, but we have updated Figure 1 and lines 123, 499 and 502 to reflect this change.

Upon further testing of our software, we have updated our code in `calculate_proportions_of_cells_in_structure()` to calculate the cell proportions in each tissue region to now include all infiltrated cells, instead of just subsets, which is more intuitive for users. This has

slightly changed the numbers reported in tables N5.1 and N5.2 in Supplementary Note N5. We have also updated the normalization of the AUC score to reflect what is described in the Methods section, line 617. The line diagram for the AUC scores of Figure 3b was also updated accordingly, but the overall shape and trends remain the same. None of these changes affect the interpretation of results or conclusions drawn, and were done to further guarantee the stability and intuitive nature of the code and our methods.

Aesthetic updates were made to figures and edits to figure legends in concordance with the guidelines of Nature Communications, as well as to accommodate the revisions requested by the reviewers.

REVIEWERS' COMMENTS

Reviewer #1 (Remarks to the Author):

The authors have answered my queries by making the package clearer for scientist without coding experience and extending their test to more tissues. I have no further comment. This package will be useful to unbiasedly analyze cell location within tissues.

Reviewer #2 (Remarks to the Author):

This reviewer has no further questions.

RESPONSE TO REVIEWERS' COMMENTS FOR MANUSCRIPT NCOMMS-22-17699A

Reviewer #1:

Reviewer comment	Response
The authors have answered my queries by making the package clearer for scientist without coding experience and extending their test to more tissues. I have no further comment. This package will be useful to unbiasedly analyze cell location within tissues.	We thank the reviewer for their support of this work.

Reviewer #2:

Reviewer comment	Response
This reviewer has no further questions.	--